# APPROXIMATION ABILITY OF TRANSFORMER NETWORKS FOR FUNCTIONS WITH VARIOUS SMOOTHNESS OF BESOV SPACES: ERROR ANALYSIS AND TOKEN EXTRACTION

## ABSTRACT

Although Transformer networks outperform various natural language processing tasks, many aspects of their theoretical nature are still unclear. On the other hand, fully connected neural networks have been extensively studied in terms of their approximation and estimation capability where the target function is included in such function classes as Hölder class and Besov class. Besov spaces play an important role in several fields such as wavelet analysis, nonparametric statistical inference and approximation theory. In this paper, we study the approximation and estimation error of Transformer networks in a setting where the target function takes a fixed-length sentence as an input and belongs to two variants of Besov spaces known as anisotropic Besov and mixed smooth Besov spaces, in which it is shown that Transformer networks can avoid curse of dimensionality. By overcoming the difficulties in limited interactions among tokens, we prove that Transformer networks can accomplish minimax optimal rate. Our result also shows that token-wise parameter sharing in Transformer networks decreases dependence of the network width on the input length. Moreover, we prove that, under suitable situations, Transformer networks dynamically select tokens to pay careful attention to. This phenomenon matches attention mechanism, on which Transformer networks are based. Our analyses strongly support the reason why Transformer networks have outperformed various natural language processing tasks from a theoretical perspective.

## 1 INTRODUCTION

Transformer networks, which were proposed in Vaswani et al. (2017), have outperformed various natural language processing (NLP) tasks, including text classifications (Shaheen et al., 2020), machine translation (Vaswani et al., 2017), language modeling (Radford et al.; Devlin et al., 2018)), and question answering (Devlin et al., 2018; Yang et al., 2019). Transformer networks make it feasible to approximate functions which can take a sequence of tokens (i.e., text) as input due to their specific architecture which is a stack of blocks of self-attention layers and token-wise feed-forward layers. However, despite of these great successes in various NLP tasks, many aspects of their theoretical nature are still unclear.

On the other hand, fully connected neural networks have been extensively studied in terms of their function approximation and estimation capability. A remarkable property of neural network is its universal approximation capability, which means that any continuous function with compact support can be approximated with arbitrary accuracy with two fully connected layers (Cybenko, 1989). However, Cybenko (1989) did not state anything about an upper bound of the network size. Therefore, a relation between properties of the target function and the network size is a next question. By imposing certain properties such as smoothness on target functions, the representabiliy of neural network can be studied more precisely. Barron (1993) developed an approximation theory for functions with limited capacity that is measured by integrability of their Fourier transform. Deep neural networks with ReLU activation (Nair & Hinton, 2010; Glorot et al., 2011) has also been extensively studied from the viewpoint of the approximation and the estimation ability. For example,

Yarotsky (2016) proved the approximation error of fully connected layers with the ReLU activation for functions in Sobolev spaces. Schmidt-Hieber (2017) derived an estimation error bound of regularized least squared estimator performed by deep ReLU network based on an approximation error analysis in a regression setting. Suzuki (2019) derived approximation and estimation error rates of fully connected layers with ReLU activation for the Besov space, which were also shown to be almost minimax optimal. Although the derived rates of convergence are almost optimal, they suffer from the curse of dimensionality, which is one of the main issues of machine learning. A typical consequence of the curse of dimensionality is that, when the dimension of data increases, the approximation accuracy (and estimation accuracy) deteriorates exponentially against the dimension. However, under some specific structure on the data and the target function, we may avoid this issue. Indeed, Suzuki (2019) and Suzuki & Nitanda (2021) showed that, by assuming that the target function has mixed smoothness or anisotropic smoothness, we can avoid curse of dimensionality. Okumoto & Suzuki (2022) derived approximation and estimation errors in a severe setting in which input data are infinite-dimesional.

Although many researches on the representation ability of fully connected layers and convolution layers are developed, relatively few researches on that of Transformer networks are found. Kratsios et al. (2021) proved that there exists a pair of an input sequence and output particles which minimize a given proper loss functions under a given constraint set. Vuckovic (2020) proved that, when regarding attention layers as functions from measures to measures, attention layers have the Lipschitz continuity property from a viewpoint of Wasserstein distances. Both Kratsios et al. (2021) and Vuckovic (2020) regard an input sentence as a measure, that is, particles or a bag of words, which is an interesting viewpoint. However, these papers do not specify how approximate Transformer networks are to a given function from an input sequence to an output. Therefore, these papers' results are different from this paper's main purpose to explain why Transformer networks can outperform various NLP tasks represented by target functions in various function spaces. Yun et al. (2020), Zaheer et al. (2020) and Shi et al. (2021) proved that Transformer networks are universal approximators of sequence to sequence functions. However, since these papers did not assume smoothness of the target function, the results of these papers did not specify an upper bound of Transformer network depths, which corresponds to the fact that universal approximation capability of neural networks did not state anything about an upper bound of the network width. Thus, this paper studies a question which naturally arises as to how properties of the target function are related to the network size and precision required.

In this paper, we study the approximation and estimation error of the Transformer architecture in a setting where the target function takes a fixed-length sentence as an input and belongs to a mixed smooth Besov space and an anisotropic Besov space. We prove that Transformer networks accomplish almost minimax optimal rate by analyzing the Transformer network architecture and approximation ability of the two function spaces. Moreover, we prove that, under suitable situations, Transformer networks can dynamically select tokens to pay careful attention to. The essence of the proof strategy is as follows: First, for a given target function, we obtain a sum of piece-wise polynomial functions which is approximate to the target function in a certain rate. Next, one constructs a neural network approximate to a piece-wise polynomial functions. Finally, one constructs a neural network approximate to the sum. The problem is the second phase in which one constructs a neural network approximate to a cardinal B-spline function. The proof of the phase is based on fully connected layers approximate to $xy$ in Yarotsky (2016). However, Transformer networks are permitted to do limited interactions among tokens. In this paper, we propose how to construct an attention layer which values exchanges between different tokens. By using attention layers constructed above, we can construct a Transformer network approximate to cardinal B-spline function. This difficulty is common to previous papers (Yun et al., 2020; Zaheer et al., 2020; Shi et al., 2021), though their strategies of obtaining a piece-wise constant approximation are different from ours in a viewpoint of exploitation of function smoothness.

Our contributions can be summarized as follows:

1. We consider a situation in which the target function takes a fixed-length sentence as an input and belongs to a mixed smooth Besov space and an anisotropic Besov space, in which it is shown that Transformer networks can avoid curse of dimentionality and accomplish almost minimax optimal rate. We also shows that token-wise parameter sharing in Transformer networks decreases dependence of the network width on the input length.

2. We prove that, under suitable situations, Transformer networks dynamically select tokens to pay careful attention to. Moreover, we show that the count of tokens to pay careful attention to is decided by an NLP task, and the accuracy required. This phenomenon matches attention mechanism, on which Transformer networks are based.

## 2   NOTATIONS AND PROBLEM SETTINGS

In this section, we define the notations and introduce the problem setting. Throughout this paper, we use the following notations. Let $X$ be a finite set. Then we shall write $\sharp X$ for the cardinality of $X$. Write $\mathbb{Z}$ for the ring of rational integers and $\mathbb{R}$ for the field of real numbers. Let $\Lambda \in \{\mathbb{Z}, \mathbb{R}\}$ and $a \in \Lambda$. Then we denote by $\Lambda_{>a} := \{a' \in \Lambda \mid a' > a\}, \Lambda_{\geq a} := \{a' \in \Lambda \mid a' \geq a\}$. Let $\Omega \subseteq \mathbb{R}^d$ be a domain of the functions. For a function $f : \Omega \to \mathbb{R}$, let $\|f\|_p := \|f\|_{L^p(\Omega)} := \left(\int_\Omega |f|^p dx\right)^{\frac{1}{p}}$ for $0 < p < \infty$ and $\|f\|_\infty := \|f\|_{L^\infty(\Omega)} := \sup_{x \in \Omega} |f(x)|$ for $p = \infty$. For $\alpha \in \mathbb{R}^d, p \in \mathbb{R}_{>0}$, we denote by $\|\alpha\|_p := \left(\sum_{i=1}^d |\alpha_i|^p\right)^{\frac{1}{p}}, \|\alpha\|_\infty := \max_{i=1}^d |\alpha_i|, \|\alpha\|_0 := \sharp\{i \in \mathbb{Z} | 1 \leq i \leq d, \alpha_i \neq 0\}$. For $\alpha \in \mathbb{R}^d_{>0}$, we denote by $\bar{\alpha} := \max(\alpha), \underline{\alpha} := \min(\alpha), \tilde{\alpha} := \left(\sum_{i=1}^d \frac{1}{\alpha_i}\right)^{-1}$. For $\alpha \in \mathbb{Z}^d_{\geq 0}$, we denote by $D^\alpha f(x) := \frac{\partial^{\|\alpha\|_1}}{\partial^{\alpha_1} x_1 \ldots \partial^{\alpha_d} x_d} f(x)$. We also define some utility functions as follows: Let $x \in \mathbb{R}$. Then we denote by $x_+ := \max(x, 0), x \vee y := \max(x, y), \lfloor x \rfloor := \min\{n \in \mathbb{Z} | n \leq x\}, \lceil x \rceil := \max\{n \in \mathbb{Z} | n \geq x\}$.

In the following subsections, the function classes for which we develop error bounds, and the set of Transformer networks with given hyper-parameters.

### 2.1   MIXED SMOOTH BESOV SPACE

In this section, we define the mixed smooth Besov space, one of the function classes which we discuss. To define the mixed smooth Besov space, we first introduce the modulus of smoothness.

**Definition 1** (*r*-th modulus of smoothness). *Let $\Omega$ be a measurable subset of $\mathbb{R}^D$, $p \in \mathbb{R}_{>0} \cup \{\infty\}$ and $r \in \mathbb{Z}_{\geq 1}$. For a function $f \in L_p(\Omega)$ and $t \in \mathbb{R}^D_{>0}$, the $r$-th modulus of smoothness of $f$ is defined by $w_{r,p}(f, t) := \sup_{|h_i| \leq t_i} \|\Delta_h^r(f)\|_p$, where $\Delta_h^r(f)(x) :=$*

$$\begin{cases} \sum_{j=0}^r \binom{r}{j}(-1)^{r-j} f(x + jh) & (x \in \Omega, x + rh \in \Omega). \\ 0 & (otherwise). \end{cases}$$

Next, based on the modulus of smoothness, we introduce the notion of mixed modulus of smoothness.

**Definition 2** (Mixed modulus of smoothness). *Let $d \in \mathbb{Z}_{\geq 1}$, $\Omega$ be a measurable subset of $\mathbb{R}^d$, $p \in \mathbb{R}_{>0} \cup \{\infty\}$, $r \in \mathbb{Z}^d_{\geq 1}$, $h \in \mathbb{R}^d$, and a function $f \in L_p(\Omega)$. Then we define the coordinate difference operator as follows: $\Delta_h^{r,i}(f)(x) := \Delta_{h_i}^r(f(x_1, \ldots, x_{i-1}, \cdot, x_{i+1}, \ldots, x_d))(x_i)$. Accordingly, for $e \subseteq \{1, \ldots, d\}$, we define the mixed difference operator $\Delta_h^{r,e}(f) := \begin{cases} \prod_{i \in e} \Delta_{h_i}^{r,i}(f) & (e \neq \emptyset) \\ f & (e = \emptyset) \end{cases}$*

*(note that, for any $i \neq j$, $\Delta_{h_i}^{r,i} \circ \Delta_{h_j}^{r,j} = \Delta_{h_j}^{r,j} \circ \Delta_{h_i}^{r,i}$) and the $r$-th mixed modulus of smoothness of $f$ $w_{r,p}^e(f, t) := \sup_{|h_i| \leq t_i} \|\Delta_h^{r,e}(f)\|_p$.*

Finally, based on the mixed modulus of smoothness, the mixed smooth Besov space is defined as in the following definition.

**Definition 3** (mixed smooth Besov space). *Let $d \in \mathbb{Z}_{\geq 1}$, $\Omega$ be a measureable subset of $\mathbb{R}^d$, $p, q \in \mathbb{R}_{>0} \cup \{\infty\}$, $\alpha \in \mathbb{R}^d_{>0}, r := \lfloor \alpha \rfloor + 1, e \subseteq \{1, \ldots, d\}$. Then, for $e \subseteq \{1, \ldots, d\}$, we define the seminorm $|\cdot|_{MB_{p,q}^{\alpha,e}}$ as follows:*

$$|f|_{MB_{p,q}^{\alpha,e}} := \begin{cases} \left(\int_\Omega ((\prod_{i \in e} t_i^{-\alpha_i}) w_{r,p}^e(f, t))^q \frac{dt}{\prod_{i \in e} t_i}\right)^{\frac{1}{q}} & (q < \infty), \\ \sup_{t \in \Omega}((\prod_{i \in e} t_i^{-\alpha_i}) w_{r,p}^e(f, t)) & (q = \infty). \end{cases}$$

Note that $|f|_{MB_{p,q}^{\alpha,\emptyset}} = ||f||_{L^p(\Omega)}$. The norm of the mixed smooth Besov space $MB_{p,q}^\alpha(\Omega)$ can be defined as the sum of the semi-norm over the choice of $e$ by $||f||_{MB_{p,q}^\alpha} := \sum_{e\subseteq\{1,...,d\}} |f|_{MB_{p,q}^{\alpha,e}}$, and we define

$$MB_{p,q}^\alpha(\Omega) := \{f \in L^p(\Omega)\big|||f||_{MB_{p,q}^\alpha} < \infty\}, MU_{p,q}^\alpha(\Omega) := \{f \in MB_{p,q}^\alpha(\Omega)\big|||f||_{MB_{p,q}^\alpha} \leq 1\}.$$

The mixed smooth Besov space was originally introduced by Schmeisser (1987); Sickel & Ullrich (2009). Various researches showed that an appropriate estimator for these models can avoid curse of dimensionality (Meier et al., 2009; Raskutti et al., 2012; Kanagawa et al., 2016; Suzuki et al., 2016). The relation between mixed smooth Besov spaces and ordinary Besov spaces, the definition of which are found in Gine & Nickl (2015); Suzuki (2019), can be informally explained as follows. A mixed smooth Besov space consists of functions for which the "maximum" of the orders of the derivatives is "bounded": $\max_{\bar{\alpha}=n} D^\alpha f$ while an ordinary Besov space consists of functions for which the "sum" of the orders of the derivatives is "bounded" $\sum_{||\alpha||=n} D^\alpha f$. This difference directly affects the rate of convergence of approximation accuracy (Düng et al. (2016); Suzuki (2019)), but, for this reason, mixed smooth Besov spaces do not include ordinary Besov spaces in general.

## 2.2 ANISOTORPIC BESOV SPACE

In this section, we define the anisotropic Besov space, the other of the function classes which we discuss. Here we define the anisotropic Besov space as follows.

**Definition 4** (Anisotropic Besov space). *Let $d \in \mathbb{Z}_{\geq 1}$, $\Omega$ be a measureable subset of $\mathbb{R}^D$, $p,q \in \mathbb{R}_{>0} \cup \{\infty\}, \alpha \in \mathbb{R}_{>0}^d, r := \max(\lfloor \alpha_i \rfloor) + 1$. Then we define the seminorm $|\cdot|_{AB_{p,q}^\alpha}$ as follows:*

$$|f|_{AB_{p,q}^\alpha} := \begin{cases} \left(\sum_{k=0}^\infty (2^k w_{r,p}(f, (2^{-\frac{k}{\alpha_1}},\ldots,2^{-\frac{k}{\alpha_d}})))^q\right)^{\frac{1}{q}} & (q < \infty), \\ \sup_{k\geq 0}(2^k w_{r,p}(f, (2^{-\frac{k}{\alpha_1}},\ldots,2^{-\frac{k}{\alpha_d}}))) & (q = \infty). \end{cases}$$

*The norm of the anisotropic Besov space $AB_{p,q}^\alpha(\Omega)$ can be defined as $||f||_{AB_{p,q}^\alpha} := ||f||_{L^p} + |f|_{AB_{p,q}^{\alpha,e}}$, and we define*

$$AB_{p,q}^\alpha(\Omega) := \{f \in L^p(\Omega)\big|||f||_{AB_{p,q}^\alpha} < \infty\}, AU_{p,q}^\alpha(\Omega) := \{f \in AB_{p,q}^\alpha(\Omega)\big|||f||_{AB_{p,q}^\alpha} \leq 1\}.$$

The statistical analysis on an anisotropic Besov space can be dated back to Ibragimov & Khas'minskii (1984), who considered an estimation of a density function which is assumed to be included in an anisotropic Sobolev space with $p \leq 2$. Afterwards, several studied have been conducted from the viewpoint of non-parametric statistics, such as nonlinear kernel estimator (Kerkyacharian et al., 2001), and kernel ridge regression (Hang & Steinwart, 2018).

Here, we present some relations with anisotropic Besov spaces and other function classes. First, if $\alpha_1 = \cdots = \alpha_d = \alpha$ then it follows from the definition of anisotropic Besov spaces that anisotropic Besov spaces are equal to oridinary Besov spaces with smoothness parameter $\alpha$. Hence, the definition of anisotropic Besov spaces includes that of the ordinary Besov spaces as a special case, while the definition of mixed smooth Besov spaces do not include that of the ordinary Besov spaces in general. Moreover, anisotropic Besov spaces are closely related to Hölder spaces. We present the definition of Hölder spaces as follows.

**Definition 5** (Hölder space). *Let $\alpha \in \mathbb{R}_{>0}$ such as $\alpha \notin N$ and we denote by $m := \lfloor \alpha \rfloor$. For an $m$ times differentiable function $f : \mathbb{R}^d \to \mathbb{R}$, let the norm of the Hölder space $\mathcal{C}_\alpha(\Omega)$ be $||f||_{\mathcal{C}_\alpha} := \max_{||n_d||_1 \leq m} ||\partial^{n_d} f||_\infty + \max_{||n_d||_1 = m} \sup_{x,y\in\Omega} \frac{|\partial^{n_d} f(x) - \partial^{n_d} f(y)|}{||x-y||^{\alpha-m}}$. Then, ($\alpha$-)Hölder space $\mathcal{C}_\alpha$ is defined as $\mathcal{C}_\alpha(\Omega) := \{f|||f||_{\mathcal{C}_\alpha} < \infty\}$.*

Let $p,q \in \mathbb{R}_{>0}^\infty$, $\alpha \in \mathbb{R}_{>0}^d$, and $\alpha_0 \in \mathbb{R}_{>0}$ such that $\tilde{\alpha} > \frac{1}{p}$. and we denote by $\alpha' := (\alpha_0,\ldots,\alpha_0)^\top$. Then, Triebel (2011) shows that $AB_{\infty,\infty}^{\alpha'} = C^{\alpha_0}, AB_{p,q}^\alpha \hookrightarrow C^0$. This result shows that, if the average smoothness $\tilde{\alpha}$ is sufficiently large ($\tilde{\alpha} > \frac{1}{p}$), then the functions in $AB_{p,q}^\alpha$ are continuous. However, it can be shown that, if it is small ($\tilde{\alpha} < \frac{1}{p}$), then they are no longer continuous. Actually, there exists functions in which spikes and jumps appear (see Donoho & Johnstone (1998) for this perspective, from the viewpoint of wavelet analysis).

## 2.3 Transformer Networks

In this section, we define the set of Transformer networks with given hyper-parameters such as fully connected layer count, transformer block count, layer width, etc. Let us denote $\mathbf{Mat}(\mathbb{R}^d, \mathbb{R}^{d'})$ by the set of linear transformations from $\mathbb{R}^d$ to $\mathbb{R}^{d'}$, for any $f : \mathbb{R}^d \mapsto \mathbb{R}^{d'}$, the function $\Pi(f) : (\mathbb{R}^d)^l \to (\mathbb{R}^{d'})^l$ by $\Pi(f)(x) := (f(x_i))_i$, for any $v, k, q \in (\mathbb{R}^d)^l$, the attention function $\mathbf{Attn} : (\mathbb{R}^d)^l \times (\mathbb{R}^d)^l \times (\mathbb{R}^d)^l \to (\mathbb{R}^d)^l$ by $\mathbf{Attn}(v, k, q) := \left( \sum_{j=1}^l v_j \frac{\exp(\langle k_j, q_i \rangle)}{\sum_{k=1}^l \exp(\langle k_k, q_i \rangle)} \right)_i$, by $\mathbf{DAttn}(x; M_K, M_Q, M_V, M_O) := M_O \cdot \mathbf{Attn}(\Pi(M_V)(x), \Pi(M_K)(x), \Pi(M_Q)(x))$, for any $P_E \in (\mathbb{R}^E)^l$, the concatenation function $\mathbf{Concat}[P_E] : (\mathbb{R}^d)^l \to (\mathbb{R}^{d+E})^l$ by $\mathbf{Concat}[P_E](x) := \begin{pmatrix} x \\ P_E \end{pmatrix}$, and the head function $\mathbf{Head} : (\mathbb{R}^d)^l \to \mathbb{R}^d$ by $\mathbf{Head}(x) := x_1$.

**Definition 6** (Transformer networks). *Define the transformer architecture* $\mathbf{TN}(L, T, E, W, H, S, B)$ *with dense layer count $L$, transformer block count $T$, width $W$, head count $H$, sparsity constraint $S$ and norm constraint $B$ recursively as follows:*
$\mathbf{FL}(W, S, B) := \{ f(x) := x + (M \cdot (x_+) + b) | M \in \mathbf{Mat}(W, W), b \in \mathbb{R}^W,$
$||M||_0 + ||b||_0 \le S, ||M||_\infty \vee ||b||_\infty \le B \},$

$\mathbf{FN}(L, W, S, B) := \{ f^{(L)} \circ \cdots \circ f^{(1)} | f^{(l)} \in \mathbf{FL}(W, S_l, B_l), \sum_{l=1}^L S_l \le S, \max_{1 \le l \le L} B_l \le B \},$

$\mathbf{AL}(W, H, S, B) := \{ f(x) := x + \sum_{h=1}^H \mathbf{DAttn}(x; M_K^{(h)}, M_Q^{(h)}, M_V^{(h)}, M_O^{(h)}) | M_s^{(h)}, M_Q^{(h)}, M_V^{(h)},$
$M_O^{(h)} \in \mathbf{Mat}(W, W), \sum_{s \in \{K, Q, V, O\}} \sum_{h=1}^H (||M_s^{(h)}||_0) \le S, \max_{\substack{1 \le h \le H \\ s \in \{K, Q, V, O\}}} (||M_s^{(h)}||_\infty) \le B \},$

$\mathbf{TL}(L, W, H, S, B) := \{ \Pi(g) \circ f | f \in \mathbf{AL}(W, H, S_1, B_1), g \in \mathbf{FN}(L, W, S_2, B_2),$
$\sum_{i=1}^2 S_i \le S, \max_{i=1,2} B_i \le B \},$

$\mathbf{STL}(L, T, W, H, S, B) := \{ f^{(T)} \circ \cdots \circ f^{(1)} | f^{(t)} \in \mathbf{TL}(L_t, W, H, S_t, B_t),$
$\sum_{t=1}^T L_t \le L, \sum_{t=1}^T S_t \le S, \max_{1 \le l \le T} B_l \le B \},$

$\mathbf{TN}(L, T, E, W, H, S, B) := \{ \mathbf{Head} \circ f \circ \mathbf{Concat}[P_E] | P_E \in (\mathbb{R}^E)^l,$
$f \in \mathbf{STL}(L, T, W, H, S_1, B), S_1 + ||P_E||_0 \le S, ||P_E||_\infty \le B \}.$

We incorporate the architecture proposed in Vaswani et al. (2017) into our definition of Transformer networks. We denote $\mathbf{FL}$ by the set of a single fully connected layer, $\mathbf{FN}$ by the set of a stack of fully connected layers, $\mathbf{AL}$ by the set of a multi-head attention layer, $\mathbf{TL}$ by the set of a Transformer block which composes of multi-head attention layer and a stack of fully connected layers, $\mathbf{STL}$ by the set of a stack of transformer blocks, $\mathbf{TN}$ by the set of an overall Transformer network with positional encoding. Note that, in order to exaggerate a count of interactions among tokens, we define a Transformer block as a concatenation of a multi-head attention layer and a stack of fully connected layers, not a single fully connected layers.

## 3 Approximation Error Analysis

In this section, we evaluate how well the functions in mixed smooth Besov and anisotropic Besov spaces can be approximated by Transformer networks. To evaluate the accuracy of the deep neural network model in approximating target functions, we first define the worst case approximation error.

**Definition 7** (Worst case approximation error). *Let $d \in \mathbb{Z}_{\ge 1}, r \in \mathbb{R}_{>0} \cup \{\infty\}$ and $\mathcal{F}, \mathcal{H}$ be subsets of measurable functions on $\Omega(\subseteq \mathbb{R}^d)$. Then we define the worst case approximation error as follows:*
$$R_r(\mathcal{F}, \mathcal{H}) := \sup_{f^\circ \in \mathcal{H}} \inf_{f \in \mathcal{F}} ||f^\circ - f||_{L^r(\Omega)}.$$

*Note that $\mathcal{F}$ is the set of functions used for approximation, and $\mathcal{H}$ is the set of target functions.*

Here, we present the results on the approximation ability.

**Theorem 1** (Approximation ability for mixed smooth Besov spaces). *Suppose that $p, q, r \in \mathbb{R}_{>0} \cup \{\infty\}, \alpha \in (\mathbb{R}^d)^l, m \in \mathbb{Z}_{\ge 1}$. Let $\delta := \left( \frac{1}{p} - \frac{1}{r} \right)_+$ (note that $\delta > 0$ is equivalent to $p < r$) and assume that $\delta < \underline{\alpha}, \bar{\alpha} < \min(m, m - 1 + \frac{1}{p})$.*

Then, for $K \in \mathbb{Z}_{\geq 1}$, there exist absolute constant $C \in \mathbb{R}_{>0}$, constants which define hyper-parameters

$$D_{k,d'} := \left(1 + \frac{d'-1}{k}\right)^k \left(1 + \frac{k}{d'-1}\right)^{d'-1}, \eta := \begin{cases} \left(\frac{1}{\min(r,1)} - \frac{1}{q}\right)_+ & (r \leq p), \\ \left(\frac{1}{r} - \frac{1}{q}\right)_+ & (p < r \text{ and } r < \infty), \\ \left(1 - \frac{1}{q}\right)_+ & (p < r \text{ and } r = \infty), \end{cases}$$

$$\nu := \frac{\alpha - \delta}{2\delta}, K^* := \left\lceil K(1 + \nu^{-1}) \right\rceil, N := (2 + (1 - 2^{-\nu})^{-1}) 2^K D_{K^*,dl},$$

$$\epsilon := 2^{-\left(\underline{\alpha} + (1+\nu^{-1})\left(\frac{1}{p} - \underline{\alpha}\right)_+\right) K}, T_0 := \lceil \log_2 l \rceil, L_1 = C \left\lceil \log\left(\frac{4dl}{\epsilon}\right) \right\rceil,$$

$$L_2 = 3 + 2 \left\lceil \log_2\left(\frac{4l \cdot 3^{d \vee m}}{\epsilon c_{(d,m)}}\right) + 5 \right\rceil \lceil \log_2(d \vee m) \rceil, L_3 = C \left\lceil \log\left(\frac{6T_0}{\epsilon}\right) \right\rceil,$$

$$W_0 := 6dm(m+2) + 2d$$

and hyper-parameters of the set of Transformer networks

$$T := T_0, E := l, H := 1, L := L_1 + L_2 + T(L_3 + 2) + 1, W := W_0 N + E,$$

$$S := C(Nl + L_1) + L_2 W_0^2 N + CT_0((N + E) + L_3) + N, B \lesssim N^{(1 + \frac{1}{\nu})\left(\left(\frac{1}{p} - \underline{\alpha}\right)_+ \vee 1\right)}.$$

such that

$$R_r(\mathbf{TN}(L, T, E, W, H, S, B), MU_{p,q}^\alpha([0,1]^{dl})) \lesssim 2^{-\underline{\alpha}K} D_{K,dl}^\eta.$$

**Theorem 2** (Approximation ability for anisotropic Besov spaces). *Suppose that $p, q, r \in \mathbb{R}_{>0} \cup \{\infty\}, \alpha \in (\mathbb{R}^d)^l, m \in \mathbb{Z}_{\geq 1}$. Let $\delta := \left(\frac{1}{p} - \frac{1}{r}\right)_+$ (note that $\delta > 0$ is equivalent to $p < r$) and assume that $\delta < \tilde{\alpha}, \bar{\alpha} < \min(m, m - 1 + \frac{1}{p})$.*

Then, for $K \in \mathbb{Z}_{\geq 1}$, there exist absolute constant $C \in \mathbb{R}_{>0}$, constants which define hyper-parameters

$$\nu := \frac{\tilde{\alpha} - \delta}{2\delta}, K^* := \left\lceil K(1 + \nu^{-1}) \right\rceil, N := (2 + (1 - 2^{-\nu})^{-1}) \tilde{N},$$

$$\epsilon := \tilde{N}^{-\left(\tilde{\alpha} + (1+\nu^{-1})\left(\frac{dl\bar{\alpha}}{p} - \underline{\alpha}\right)_+\right)}, T_0 := \lceil \log_2 l \rceil, L_1 = C \left\lceil \log\left(\frac{4dl}{\epsilon}\right) \right\rceil,$$

$$L_2 = 3 + 2 \left\lceil \log_2\left(\frac{4l \cdot 3^{d \vee m}}{\epsilon c_{(d,m)}}\right) + 5 \right\rceil \lceil \log_2(d \vee m) \rceil, L_3 = C \left\lceil \log\left(\frac{6T_0}{\epsilon}\right) \right\rceil,$$

$$W_0 := 6dm(m+2) + 2d$$

and hyper-parameters of the set of Transformer networks

$$T := T_0, E := l, H := 1, L := L_1 + L_2 + T(L_3 + 2) + 1, W := W_0 N + E,$$

$$S := C(Nl + L_1) + L_2 W_0^2 N + CT_0((N + E) + L_3) + N, B \lesssim \tilde{N}^{(1 + \nu^{-1})\left(\left(\frac{dl\bar{\alpha}}{p} - \underline{\alpha}\right)_+ \vee \bar{\alpha}\right)}.$$

such that

$$R_r(\mathbf{TN}(L, T, E, W, H, S, B), AU_{p,q}^\alpha([0,1]^{dl})) \lesssim \tilde{N}^{-\tilde{\alpha}}.$$

The proofs of these theorems are provided in Appendix D. Note that the upper bound in the inequality of Theorem 1 depends only on $D_{K,dl}$ ($D_{K,dl}$ mildly depends on $d$ and $l$) and the upper bound in that of Theorem 2 does not depend on $d$ or $l$. This means that if the target function is included these function classes, we can ease the curse of dimensionality. Moreover, thanks to the token-wise parameter sharing property in Transformer networks, the width of the network architecture does not depend on the input length $l$, but only on the feature dimension $d$. Thus, our result also shows that the extent to which the network width and the approximation error upper bounds depend on the input length can be relaxed. Hence, Transformer networks are more efficient in a network size than fully-connected layers (See Suzuki (2019) and Suzuki & Nitanda (2021)).

**Remark 1.** *We give the following approximation bound by using an adaptive sampling recovery method developed by (Dũng, 2011a). The key point of this technique is that, instead of the whole set of the basis functions, we adaptively select much smaller functions from the whole set to approximate functions. If the target function belongs to mixed smooth or anisotropic Besov spaces, we can use this adaptive technique. Therefore, we deal with the variants of Besov spaces in this paper.*

## 4 ESTIMATION ERROR ANALYSIS

In this section, we connect the approximation theory to estimation error analysis. First, we define the settings in the non-parametric regression model:

**Definition 8** (Non-parametric regression model for statistical analysis). *Let $f^\circ : ([0,1]^d)^l \to \mathbb{R}$ be a measurable function and*

$$y_i := f^\circ(x_i) + \xi_i$$

*where $x_i \sim P_X$ with density function $0 \leq p(x) < R$ on $([0,1]^d)^l$, and $\xi_i \sim N(0, \sigma^2)$. We denote the training data $D_n := (x_i, y_i)_{i=1}^n$ which are independently identically distributed. Here, we define a regularized learning estimator as follows:*

$$\hat{f} := \underset{\bar{f}:f\in\Phi(L,T,E,W,H,S,B)}{\operatorname{argmax}} \sum_{i=1}^n (y_i - \bar{f}(x_i))^2$$

*where $\bar{f}$ is the clipping of $f$ defined by $\bar{f} := \min\{\max\{f, -F\}, F\}$ for $F > 0$ which is realized by ReLU units.*

In practice, it is hard to exactly compute $\hat{f}$ in the definition above. Therefore, there are numerous researches which study how to approximately compute $\hat{f}$ by applying sparse regularization such as $L^1$ regularization and optimal parameter search through Bayesian optimization. In this study, we assume that the optimal solution $\hat{f}$ is computable. Thus, we can assume that $\hat{f}$ in the definition above is valid.

Here, we provide the estimation error rate of deep learning to estimate functions in Besov spaces by using the approximation error bound given in the previous sections.

**Theorem 3.** *Suppose that $p, q \in \mathbb{R}_{>0} \cup \{\infty\}, \alpha \in (\mathbb{R}_{>0}^d)^l$. If $f^\circ \in MB_{p,q}^\alpha \cap L^\infty(\Omega)$ and $||f^\circ||_{MB_{p,q}^\alpha} \leq 1$ and $||f^\circ||_{L^\infty} \leq F$, then letting $(L, T, E, W, H, S, B)$ be as in Theorem 1, we obtain*

$$\mathbb{E}_{D_n}\left(||f^\circ - \hat{f}||_{L^2(P_X)}\right) \lesssim n^{-\frac{2\alpha}{2\alpha+1}} \log(n)^{\frac{2(dl-1)(\eta+\alpha)+6\alpha}{1+2\alpha}}$$

*where $\eta := \eta_{p,q,r}$ as in the notation of Theorem 1.*

**Theorem 4.** *Suppose that $p, q \in \mathbb{R}_{>0} \cup \{\infty\}, \alpha \in (\mathbb{R}_{>0}^d)^l$. If $f^\circ \in AB_{p,q}^\alpha \cap L^\infty(\Omega)$ and $||f^\circ||_{AB_{p,q}^\alpha} \leq 1$ and $||f^\circ||_{L^\infty} \leq F$, then letting $(L, T, E, W, H, S, B)$ be as in Theorem 1, we obtain*

$$\mathbb{E}_{D_n}\left(||f^\circ - \hat{f}||_{L^2(P_X)}\right) \lesssim n^{-\frac{2\tilde{\alpha}}{2\tilde{\alpha}+1}} \log(n)^{\frac{6\tilde{\alpha}}{1+2\tilde{\alpha}}}.$$

The proofs are given in Appendix D. The condition $||f^\circ||_\infty \leq F$ is used to fill a gap between the empirical $L^2$-norm and the population $L^2$-norm. A key factor of these results is a fact that the complexity of Transformer networks is not so high as that of fully connected layers to some extent. By combining this fact and the approximation error analysis in Section 3, the above estimations follows. Note that the dimensional parameters $d, l$ do not appear in the exponent of $n$ in the upper bounds, but only in the exponent of $\log(n)$ term. Thus, the risk bound (Theorem 3 and Theorem 4) indicates that curse of dimensionality can be relaxed in the two variants of Besov spaces. We can see that there does not appear $d, l$ directly in the exponent of the convergence rate (although it appears in the poly-log term for the mixed smooth case). Instead, the rate is mainly characterized by $\underline{\alpha}, \tilde{\alpha}$. This means that the curse of dimensionality is eased by utilizing the smoothness structure of the true function $f^\circ$.

**Remark 2.** *According to Suzuki (2019) and Suzuki & Nitanda (2021), $\inf_{\hat{f}} \sup_{f^\circ \in U} \mathbb{E}_{D_n}\left(||f^\circ - \hat{f}||_{L^2(P_X)}\right) \gtrsim n^{-\frac{2\tilde{\alpha}}{2\tilde{\alpha}+1}} \log(n)^{\frac{6\tilde{\alpha}}{1+2\tilde{\alpha}}}$ in the case of mixed smooth Besov spaces and anisotropic Besov spaces holds. Thus, by combining Theorem 3 and 4, we show that Transformer networks accomplish almost minimax optimal rate up to a poly-$\log(n)$ order, and, especially, under the conditions of $p < 2$ and $1/2 - 1/q > 0$, accomplish almost minimax optimal rate up to $\log(n)^3$ order.*

*Thus, Transformer networks have the potential to best fit the target function in either mixed smooth Besov spaces or anisotropic Besov spaces among all estimators. Note that it has been already shown*

*in Suzuki (2019) and Suzuki & Nitanda (2021) that fully connected layers achieve almost minimax optimal rate. Therefore, by Definition 6, it is intuitively true that Transformer networks also achieve almost minimax optimal rate. The important thing is that we prove that Transformer networks are more efficient than fully connected layers in a setting where this intuition is true. For an instance, the extent to which the network width and the approximation error upper bounds depend on the input length can be relaxed, as we show in Section 3.*

## 5    TOKEN EXTRACTION

In this section, we discuss the token extraction property of Transformer networks. First, we introduce a new function class which is a variant of mixed smooth Besov spaces to express a situation in which Transformer networks dynamically select important tokens for an input sequence.

**Definition 9.** *Let $\Omega, \Omega_i \subseteq \mathbb{R}^d$ and $\Omega = \bigsqcup_{i=1}^n \Omega_i$, and $\alpha_i \in \mathbb{Z}_{\geq 0}^d$ where $\Omega, \Omega_i$ are written as $\Pi_{i=1}^d I_i$ when $I_i := [a_i, b_i], [a_i, b_i), (a_i, b_i],$ or $(a_i, b_i)$. We denote a partition of $\Omega$ as $\pi := (\Omega_i)$, and piecewise smoothness as $\alpha := (\alpha_i)$. Then, the norm of the variable mixed smooth Besov space $VB_{p,q}^{\alpha,\pi}(\Omega)$ can be defined as follows:*

$$\|f\|_{VB_{p,q}^{\alpha,\pi}} := \sum_{i=1}^n \|f\|_{MB_{p,q}^{\alpha_i}(\Omega_i)},$$

*and we define*

$$VB_{p,q}^{\alpha,\pi}(\Omega) := \left\{ f \in L^p(\Omega) \big| \|f\|_{VB_{p,q}^{\alpha,\pi}} < \infty \right\}, VU_{p,q}^{\alpha}(\Omega) := \left\{ f \in VB_{p,q}^{\alpha}(\Omega) \big| \|f\|_{VB_{p,q}^{\alpha}} \leq 1 \right\}.$$

Intuitively, the target function in the variable mixed smooth Besov space changes a direction to regard as important or as a noise, according to an input. For each region $\Omega_i$, a corresponding smoothness parameter $\alpha_i$ decides whether a direction is important or a noise. By regarding each direction as a token, we can express that the target function decide which tokens to pay attention to, for an input sequence (or a set of input sequences).

Next, we introduce input quantization masks. Input quantization masks are used to cut off information of masked tokens to specify that Transformer networks extract much more information from non-masked tokens and much less information from masked tokens.

**Definition 10.** *Let $t, u \in \mathbb{Z}_{\geq 1}$. Then we denote as $\mathcal{Q}_{t,u}$ the set of input quantization masks $f$ as follows: If there exist a partition $\pi := (\Omega_k)$ and subsets $S_k \subseteq \{1, 2, \ldots, l\}$ such that $\sharp S_k \leq t$,*

$$f(x) := (x_{ij}\mathbb{I}(x, j)) \text{ where } \mathbb{I}(x, j) := \begin{cases} x_{ij} & (x \in \Omega_k, j \in S_k) \\ \frac{\lceil x_{ij} u \rceil}{u} & (x \in \Omega_k, j \notin S_k) \end{cases}. \text{ By using the definition of the}$$

*set of input masks, we define the set of transformer networks with input mask as follows:*

$$\mathbf{MTN}_{t,u}(L, T, E, W, H, S, B) := \left\{ f \circ q \big| f \in \mathbf{TN}(L, T, E, W, H, S, B), q \in \mathcal{Q}_{t,u} \right\}.$$

Intuitively, masked tokens have much less information than non-masked tokens because masked tokens are rounded up by multiples of $u$. For example, when $u = 2$, $x_{ij} \in (0, 1/2]$ are rounded up to $1/2$, and $x_{ij} \in (1/2, 1]$ are rounded up to $1$. Hence, we see that this round-up quantization cut off much information of original tokens. Note that the higher the parameter $u$ is, more roughly an input value is rounded up (or quantized). The parameter $u$ in the definition is needed for a certain technical reason. The reason is why we use cardinal B-splines (which is not a constant function but a piecewise polynomial) to approximate the target function in this paper.

By using the definitions above, we can present the main result in this section.

**Theorem 5** (Token extraction property of Transformer networks). *Let $s \in \mathbb{R}_{>1 \vee \frac{1}{p}}, \pi := (\Omega_k), \alpha := (\alpha_k), \sigma := (\sigma_k)$ where $\alpha_k \in (\mathbb{R}_{>0}^d)^l$ and $\sigma_k$ be a permutation on $\{1, 2, \ldots, l\}$. Moreover, we assume that $r \geq 1$ and $(\alpha_k)_{ij} \geq s^{\sigma_k(j)-1}$. Then, for $K \in \mathbb{Z}_{\geq 1}$, letting $(L, T, E, W, H, S, B)$ be as in Theorem 1, there exists constants $t := \left\lceil \frac{\log(\frac{1}{p} + K)}{\log s} \right\rceil$ and $u := 2^K$ such that a following estimation holds:*

$$R_r(\mathbf{MTN}_{t,u}(L, T, E, W, H, S, B), VU_{p,q}^{\alpha}([0, 1]^{dl})) \lesssim 2^{-K} D_{K,dl}^{\eta \vee 1}.$$

The proof is given in Appendix E. First, we explain the role of variable mixed smooth Besov spaces. Variable mixed smooth Besov spaces can be regarded as the set of target functions which, for an input sequence (or a set of input sequences) decide which tokens to pay attention to. Smoothness parameters $\alpha_i \in (\mathbb{R}^d)^l$ have control over this token selection process. For example, let us consider a text classification task. When classifying a text "FRB is stepping up its battle on inflation" to a finance category, we will pay attention to the first word "FRB" and the eighth word "inflation", and, when classifying a text "Moreover, borrowing costs are going sharply higher" to a finance category, we will pay attention to the second word "borrowing" and the third word "inflation". Therefore, variable mixed smooth Besov spaces can grasp how a task decide which tokens to pay attention to. Under the settings of Theorem 5, the smoothness parameters $(\alpha_k)_{ij}$ increases with respect to $\sigma(j)$ in an exponential order. This means that the *importance* of token $j$ by token $i$ exponentially decays under an appropriate permutation $\sigma_k$ which can depend on the input $x$. Then, Theorem 5 shows that the Transformer can detect the *input-dependent importance* between tokens and achieves the addaptive rate which cannot be obtained by imposing a fixed smoothness over the entire input $x$.

Next, we explain the role of input quantization masks. Note that the range of masked token feature values is $\{0, \frac{1}{u}, \frac{2}{u}, \ldots, 1 - \frac{1}{u}, 1\}$, while the range of non-masked token features is $[0, 1]$. Thus, the cardinality of the range of masked token feature values is finite while the cardinality of the range of non-masked token feature values are uncountably infinite. Hence, masked tokens have much less information than non-masked tokens. Thus, input masks expresses a situation in which Transformer networks extract much more information from non-masked tokens and much less information from masked tokens. As mentioned above, the parameter $u$ in the definition is needed for a certain technical reason. In this paper, we use cardinal B-splines (which is not a constant function but a piecewise polynomial) to approximate the target function, and, since piecewise constant functions are needed to approximate cardinal B-splines, input masks need quantization. Actually, since Okumoto & Suzuki (2022) considered a space of functions which are (possibly infinite) sums of finite products of trigonometric functions, this technical problem did not occur in Okumoto & Suzuki (2022).

Consequently, $\mathbf{MTN}_t$ in Definition 10 can be regarded as the set of transformer networks which, for an input sequence (or a set of input sequences), fully exploit at most $t$ tokens' features of an input sequence.

Thus, Theorem 5 shows that, for a general NLP task and accuracy required, Transformer networks can dynamically select $t$ tokens to pay careful attention to (a value $t$ is decided by the target function which represents the NLP task and the accuracy required). This token selection property matches attention mechanism, on which Transformer networks are based.

## 6 CONCLUSION

This paper investigated the learning ability of Transformer networks when the target function is in mixed smooth Besov spaces or anisotropic Besov spaces. By overcoming the difficulties in limited interactions among tokens, we show that Transformer networks can adaptively avoid curse of dimensionality and accomplish minimax optimal rate. Our result also shows that dependence of the network width on the input length and the approximation error upper bounds can be relaxed, thanks to token-wise parameter sharing in Transformer networks. Moreover, we prove that, when the smoothness parameters $\alpha_{ij}$ increases in an exponential order of a permutation of the token location $j$, Transformer networks dynamically select tokens to pay careful attention to. This phenomenon matches attention mechanism, and the result suggests that this favorable property is derived to the architecture of Transformer networks. Our analyses strongly support the reason why Transformer networks have outperformed various natural language processing tasks from a theoretical perspective.

This paper did not discuss the optimization aspect of networks. In this paper, we assume that the optimal solution of regularized least squares are computable. For future works, it would be interesting to incorporate non-convex optimization techniques into our study.

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

## A    DEFINITION AND VARIOUS RESULTS ON THE CARDINAL B-SPLINE

Here, we define the cardinal B-spline and present auxiliary lemmas. If the target function belongs to various Besov spaces (DeVore & Popov, 1988; DeVore et al., 1993; Dũng, 2011a). we can obtain its B-spline interpolant representations. Thus, we can see that an approximation of a function in various Besov spaces can reduce to an approximation of the cardinal B-spline.

**Definition 11** (Cardinal B-spline). *We define the cardinal B-spline of order $m$ as follows:*

$$\mathcal{N}(x) := \begin{cases} 1 & (x \in [0,1]) \\ 0 & (x \notin [0,1]) \end{cases}$$

*and $\mathcal{N}_0(x) := \mathcal{N}(x)$, $\mathcal{N}_{m+1}(x) := \mathcal{N}_m(x) * \mathcal{N}(x)$ where $(f * g)(x) := \int_{\mathbb{R}} f(x-t)g(t)dt$ is a convolution of $f$ and $g$.*

*Let $d \in \mathbb{Z}_{\geq 1}, k, j \in \mathbb{Z}_{\geq 0}^d$. Then we define*

$$M_{0,0}^d(x) := \prod_{i=1}^d N_m(x_i),$$

$$M_{k,j}^d(x) := M_{0,0}^d(2^{k_i} x_i - j_i).$$

$$J_m^d(k) := \{-m, -(m-1), \dots, 2^{k_1} - 1, 2^{k_1}\} \times \cdots \times \{-m, -(m-1), \dots, 2^{k_d} - 1, 2^{k_d}\}.$$

**Lemma 1** (Property of B-splines). $\|\mathcal{N}_m\|_{L^\infty} \leq 1$ *and, if $m \geq 1$, $\mathcal{N}_m$ are 1-Lipschitz.* $\left\|M_{0,0}^d\right\|_{L^\infty} \leq 1$ *and, if $m \geq 1$, $M_{0,0}^d$ are $d$-Lipschitz.*

*Proof.* If $m = 0$, it clearly follows from the definition of $N$ that $\|\mathcal{N}_m\|_{L^\infty} \leq 1$. If $m \geq 1$, it follows that

$$\|\mathcal{N}_m\|_{L^\infty} = \sup_{x \in \mathbb{R}} \left| \int_{\mathbb{R}} \mathcal{N}_{m-1}(x-t)\mathcal{N}(t)dt \right| \leq \sup_{x \in \mathbb{R}} \left| \int_{\mathbb{R}} \mathcal{N}(t)dt \right| = 1,$$

and

$$|\mathcal{N}_m(x_1) - \mathcal{N}_m(x_2)| = \left| \int_{\mathbb{R}} (\mathcal{N}_{m-1}(x_1 - t) - \mathcal{N}_{m-1}(x_2 - t))\mathcal{N}(t)dt \right|$$

$$\leq |x_1 - x_2| \cdot \left| \int_{\mathbb{R}} \mathcal{N}(t)dt \right| = |x_1 - x_2|.$$

Thus, it clearly follows $\left\|M_{0,0}^d\right\|_{L^\infty} \leq 1$ and

$$\left| M_{0,0}^d(x_1) - M_{0,0}^d(x_2) \right|_{L^\infty}$$

$$= \sum_{1 \leq i \leq d} \left| \prod_{1 \leq j \leq i-1} \mathcal{N}_m((x_1)_i) \right| |\mathcal{N}_m((x_1)_i) - \mathcal{N}_m((x_2)_i)| \left| \prod_{i+1 \leq j \leq d} \mathcal{N}_m((x_2)_i) \right|$$

$$\leq \sum_{1 \leq i \leq d} |(x_1)_i - (x_2)_i| \leq d \|x_1 - x_2\|_\infty.$$

$\square$

**Lemma 2** ($L^p$ norm of a linear combination of B-splines). *Let $d \in \mathbb{Z}_{\geq 1}$, $k \in \mathbb{Z}_{\geq 0}^d$, and $f := \sum_{j \in J_m^d(k)} c_j M_{k,j}^d$ a linear combination of B-splines. Then we estimate the $L^p$ norm of the linear combination $f$ as follows:*

$$||f||_{L^p} \simeq 2^{-\frac{||k||_1}{p}} \left( \sum_{j \in J_m^d(k)} |c_j|^p \right)^{\frac{1}{p}}$$

The proof is found in Suzuki (2019).

## B    SUB-NETWORKS

Here, we present auxiliary lemmas, which are used to sub-networks of which Transformer networks compose. A key step to show the approximation accuracy is to construct a ReLU neural network which can approximate the cardinal B-spline with high accuracy. By using the technique developed by Yarotsky (2016), we can construct fully connected layers with ReLU activation functions to approximate the cardinal B-spline. By combining the result B-spline approximation results and the results in this section, we can obtain the optimal approximation error bound for Transformer networks.

**Lemma 3** (Approximation of $x^2$). *Let $\epsilon \in \mathbb{R}_{>0}$. Then, there exist constants*

$$L_1 := \left\lceil \log_2 \left( \frac{1}{\epsilon} \right) \right\rceil, W_1 := 4, S_1 := 8 \left\lceil \log_2 \left( \frac{1}{\epsilon} \right) \right\rceil, B_1 := 1,$$

*and a neural network $M_1 \in \Phi_2(L_1, W_1, S_1, B_1)$.such that*

$$\sup_{x \in [0,1]} \left| x^2 - M_1(x) \right| \leq \epsilon.$$

*Moreover, if $R \in \mathbb{R}_{\geq 1}$, then, there exist constants*

$$L_2 := \left\lceil \log_2 \left( \frac{R^2}{\epsilon} \right) \right\rceil + 3, W_2 := 4, S_2 := 8 \left\lceil \log_2 \left( \frac{R^2}{\epsilon} \right) \right\rceil + 3, B_2 := R,$$

*and a neural network $M_2 \in \Phi_2(L, W, S, B)$.such that*

$$\sup_{x \in [0,R]} \left| x^2 - M_2(x) \right| \leq \epsilon.$$

*Proof.* If $R = 1$, the proof is found in Proposition 2, Yarotsky (2016). If $R > 1$, we can obtain the following network:

$$x \xrightarrow{\times \frac{1}{R^2}} \cdot \xrightarrow{M_1} \cdot \xrightarrow{\times R} \cdot \xrightarrow{\times R} \cdot.$$

$\square$

**Lemma 4** (Approximation of $xy$). *Let $\epsilon, R \in \mathbb{R}_{\geq 1}$. Then, there exist constants*

$$L := \left\lceil \log \left( \frac{6R^2}{\epsilon} \right) \right\rceil + 5, W := 12, S := 24 \left\lceil \log \left( \frac{6R^2}{\epsilon} \right) \right\rceil + 14, B := R,$$

*and a neural network $M \in \Phi_2(L, W, S, B)$.such that*

$$\sup_{x \in [0,R]} |xy - M((x,y))| \leq \epsilon.$$

*Proof.* The proof strategy is found in Proposition 3, Yarotsky (2016). $\square$

**Lemma 5** (Approximation of cardinal B-spline basis by the ReLU activation). *Let $d \in \mathbb{Z}_{\geq 1}$. Then, there exists a constant $c(d, m)$ depending only on $d$ and $m$ such that, for all $\epsilon > 0$, there exist constants*

$$L_0 := 3 + 2 \left\lceil \log_2 \left( \frac{3^{d \vee m}}{\epsilon c_{(d,m)}} \right) + 5 \right\rceil \lceil \log_2(d \vee m) \rceil, W_0 := 6dm(m+2) + 2d,$$

$$S_0 := L_0 W_0^2, B_0 := 2(m+1)^m,$$

*and a neural network $M \in \Phi_2(L, W, S, B)$.such that*

$$||M_{0,0}^d - M||_{L^\infty(\mathbb{R}^d)} \leq \epsilon$$

*and $M(x) = 0 \ (\forall x \notin [0, m+1]^d)$.*

*Proof.* The proof is found in Lemma 1, Suzuki (2019). $\square$

## C   PROOF OF THE STATEMENTS OF SECTION 3

Here, we give technical details behind the approximation bound. We will use the so-called sparse grid technique which Smolyak (1963) introduced to the function approximation theory field. The key point of this technique is that, instead of the whole set of the regular grid, we put the basis on a sparse grid which is a subset of the whole set and has much smaller cardinality than the whole set. The applications of approximation algorithm were developed by Düng (1990; 1991; 1992); Temlyakov (1982; 1993a;b). Afterwards, the sparse grid technique develops into an optimal adaptive sampling recovery method by (Dũng, 2011b), and we adopt this method on the cardinal B-spline bases. We follow the proof strategy for Suzuki (2019) and Suzuki & Nitanda (2021).

**Definition 12.** *Let $d \in \mathbb{Z}_{\geq 1}$, $p_k \in \mathbb{R}$ for $k \in \mathbb{Z}_{\geq 0}^d$, and $c_{k,j} \in \mathbb{R}$ for $k \in \mathbb{Z}_{\geq 0}^d, j \in J_m^d(k)$. Then we define a quasi-norm over a set of functions by*

$$|(p_k)|_{b_q^\alpha(L^p)} := \left( \left[ \sum_{k \in \mathbb{Z}_{\geq 0}^d} (2^{\langle \alpha, k \rangle} \|p_k\|_p) \right]^q \right)^{\frac{1}{q}},$$

*and a quasi-norm over a set of coefficients by*

$$|(c_{k,j})|_{mb_{p,q}^\alpha} := \left( \left[ \sum_{k \in \mathbb{Z}_{\geq 0}^d} \left( 2^{\langle \alpha, k \rangle - \frac{\|k\|}{p}} \left( \sum_{j \in J_m^d(k)} |c_{k,j}|^p \right)^{\frac{1}{p}} \right)^q \right] \right)^{\frac{1}{q}}.$$

**Theorem 6** (Cardinal B-spline approximation for mixed smooth Besov spaces). *Suppose that $p, q, r \in \mathbb{R}_{>0} \cup \{\infty\}, \alpha \in \mathbb{R}_{>0}^d$. Let $\delta := \left( \frac{1}{p} - \frac{1}{r} \right)_+$ (note that $\delta > 0$ is equivalent to $p < r$) and assume that $m \in \mathbb{Z}_{\geq 1}$, and $\delta < \underline{\alpha}, \bar{\alpha} < \min \left( m, m - 1 + \frac{1}{p} \right).$*

*Then, for any $f \in MB_{p,q}^\alpha$ and $K \in \mathbb{Z}_{\geq 1}$, there exist constants*

$$\eta := \begin{cases} (\frac{1}{\min(r,1)} - \frac{1}{q})_+ & (r \leq p), \\ (\frac{1}{r} - \frac{1}{q})_+ & (p < r \text{ and } r < \infty), \\ (1 - \frac{1}{q})_+ & (p < r \text{ and } r = \infty), \end{cases}$$

$$\nu := \frac{\alpha - \delta}{2\delta}, K^* := \lceil K (1 + \nu^{-1}) \rceil, n_k := \left\lceil 2^{K - \nu(\|k\| - K)} \right\rceil,$$

$$S(k) \subseteq J_m^d(k) \text{ such that } \sharp(S(k)) = n_k,$$

*and*

$$R_K(f) := \sum_{\substack{k \in \mathbb{Z}_{\geq 0} \\ \|k\|_1 \leq K}} \sum_{j \in J_m^d(k)} c_{k,j} M_{k,j}^d(x) + \sum_{\substack{k \in \mathbb{Z}_{\geq 0} \\ K < \|k\|_1 \leq K^*}} \sum_{j \in S(k)} c_{k,j} M_{k,j}^d(x)$$

*such that*

$$\|f - R_K(f)\|_r \lesssim 2^{-\underline{\alpha}K} D_{K,d}^{\eta_{p,q},r} \|f\|_{MB_{p,q}^\alpha},$$

$$\sharp E(K) := \{(k,j) \in \mathbb{Z}_{\geq 0}^d \times \mathbb{Z}_{\geq 0}^d | c_{k,j} \neq 0\} \leq \left(2 + (1 - 2^{-\nu})^{-1}\right) 2^K D_{K^*,d},$$

$$k_{\max} := \max_{\substack{1 \leq i \leq d \\ k \in E_K}} k_i \leq K^*, c_{\max} := \max_{\substack{1 \leq i \leq d \\ k \in E_K}} c_{k,j} \lesssim 2^{K^*\left(\frac{1}{p} - \alpha\right)_+} \|f\|_{MB_{p,q}^\alpha}.$$

*Proof of Theorem 6.* According to Suzuki (2019) (see also Dũng (2011a)), there exist a collection of functions $(\mathcal{P}_k)_{k \in \mathbb{Z}_{\geq 0}^d}$ from $MB_{p,q}^\alpha$ to $MB_{p,q}^\alpha$ such that

$$\|f\|_{MB_{p,q}^\alpha} \simeq \|(p_k)\|_{b_q^\alpha(L^p)} \simeq \|(c_{k,j})\|_{mb_{p,q}^\alpha}$$

where $p_k := \mathcal{P}_k(f) = \sum_{j \in J_m^d(k)} c_{k,j} M_{k,j}^d(x)$.

**(1) the case of $r \leq p$.** Then, the assertion can be shown in the same manner as Theorem 3.1 of Dũng (2011a).

**(2) the case of $p < r$.** We need to use an adaptive approximation method. In the following, we assume $p < r$. For a given $K$, by choosing $K^*$ appropriately later, we set

$$R_K(f) := \sum_{\substack{k \in \mathbb{Z}_{\geq 0} \\ \|k\|_1 \leq K}} p_k + \sum_{\substack{k \in \mathbb{Z}_{\geq 0} \\ K < \|k\|_1 \leq K^*}} G_k(p_k)$$

where $G_k(p_k)$ is given as

$$G_k(p_k) := \sum_{i=1}^{n_k} c_{k,j_i} M_{k,j}^d(x)$$

where $(c_{k,j_i})$ is the sorted coefficients in decreasing order of their absolute value: $|c_{k,j_1}| \geq |c_{k,j_1}| \geq \cdots \geq |c_{k,\sharp J_m^d(k)}|$.

$$R_K(f) := \sum_{\substack{k \in \mathbb{Z}_{\geq 0} \\ \|k\|_1 \leq K}} \sum_{j \in J_m^d(k)} c_{k,j} M_{k,j}^d(x) + \sum_{\substack{k \in \mathbb{Z}_{\geq 0} \\ K < \|k\|_1 \leq K^*}} \sum_{j \in J_m^d(k)} c_{k,j} M_{k,j}^d(x)$$

Then, it holds that

$$\|p_k - G_k(p_k)\|_r \leq \|p_k\|_p 2^{\delta \|k\|_1} n_k^{-\delta},$$

where $\delta := (1/p - 1/r)$ (see also the proof of Dũng (2011a) and Dũng (2011b)).

Here we denote by

$$\nu := \frac{\alpha - \delta}{2\delta}, K^* := \left\lceil K\left(1 + \nu^{-1}\right)\right\rceil, n_k := \left\lceil 2^{K - \nu(\|k\| - K)}\right\rceil.$$

Then, by Lemma 5.3 of Dũng (2011a), it follows that

$$\|f - R_K(f)\|_{L^r}^r \lesssim \sum_{\substack{k \in \mathbb{Z}_{\geq 0} \\ K < \|k\|_1 \leq K^*}} \left(2^{\langle \delta \mathbb{1}, k \rangle} n_k^{-\delta} \|p_k\|_{L^p}\right)^r + \sum_{\substack{k \in \mathbb{Z}_{\geq 0} \\ K^* < \|k\|_1}} \left(2^{\langle \delta \mathbb{1}, k \rangle} \|p_k\|_{L^p}\right)^r.$$

**(2-1) the case of $r < \infty$ and $q \leq r$**

$$||f - R_K(f)||_{L^r}^q$$

$$\lesssim \left( \sum_{\substack{k \in \mathbb{Z}_{\geq 0} \\ K < ||k||_1 \leq K^*}} \left( 2^{\langle \delta \mathbb{1}, k \rangle} n_k^{-\delta} ||p_k||_{L^p} \right)^r + \sum_{\substack{k \in \mathbb{Z}_{\geq 0} \\ K^* < ||\bar{k}||_1}} \left( 2^{\langle \delta \mathbb{1}, k \rangle} ||p_k||_{L^p} \right)^r \right)^{\frac{q}{r}}$$

$$\leq \sum_{\substack{k \in \mathbb{Z}_{\geq 0} \\ K < ||k||_1 \leq K^*}} \left( 2^{\langle \delta \mathbb{1}, k \rangle} n_k^{-\delta} ||p_k||_{L^p} \right)^q + \sum_{\substack{k \in \mathbb{Z}_{\geq 0} \\ K^* < ||\bar{k}||_1}} \left( 2^{\langle \delta \mathbb{1}, k \rangle} ||p_k||_{L^p} \right)^q \left( \text{since } \frac{q}{r} \leq 1 \right)$$

$$\leq 2^{-\delta K q} 2^{-(\underline{\alpha} - \delta) K q} \sum_{\substack{k \in \mathbb{Z}_{\geq 0} \\ K < ||k||_1 \leq K^*}} \left( 2^{-(\underline{\alpha} - \delta - \delta\nu)(||k||_1 - K)} 2^{\langle \alpha, k \rangle} ||p_k||_{L^p} \right)^q$$

$$+ 2^{-q(\underline{\alpha} - \delta) K^*} \sum_{\substack{k \in \mathbb{Z}_{\geq 0} \\ K^* < ||\bar{k}||_1}} \left( 2^{\langle \alpha, k \rangle} ||p_k||_{L^p} \right)^q$$

$$\lesssim 2^{-\underline{\alpha} K q} ||f||_{MB_{p,q}^\alpha}.$$

**(2-2) the case of $r < \infty$ and $q > r$**

Since $\frac{1}{\frac{r}{q}} + \frac{1}{\frac{q-r}{q}} = 1$, then it follows by applying Hölder's inequality that

$$||f - R_K(f)||_{L^r}^r$$

$$\lesssim \sum_{\substack{k \in \mathbb{Z}_{\geq 0} \\ K < ||k||_1 \leq K^*}} \left( 2^{\delta ||k||_1} n_k^{-\delta} ||p_k||_{L^p} \right)^r + \sum_{\substack{k \in \mathbb{Z}_{\geq 0} \\ K^* < ||\bar{k}||_1}} \left( 2^{\delta ||k||_1} ||p_k||_{L^p} \right)^r$$

$$\lesssim 2^{-\underline{\alpha} K r} \left( \sum_{\substack{k \in \mathbb{Z}_{\geq 0} \\ K < ||k||_1 \leq K^*}} \left( 2^{-(\underline{\alpha} - \delta - \delta\nu)(||k||_1 - K)} 2^{\langle \alpha, k \rangle} ||p_k||_{L^p} \right)^r \right.$$

$$\left. + \sum_{\substack{k \in \mathbb{Z}_{\geq 0} \\ K^* < ||\bar{k}||_1}} \left( 2^{-(\underline{\alpha} - \delta)(||k||_1 - K^*)} 2^{\langle \alpha, k \rangle} ||p_k||_{L^p} \right)^r \right)$$

$$\leq 2^{-\underline{\alpha} K r} \left( \sum_{\substack{k \in \mathbb{Z}_{\geq 0} \\ K < ||k||_1 \leq K^*}} \left( 2^{\langle \alpha, k \rangle} ||p_k||_{L^p} \right)^q + \sum_{\substack{k \in \mathbb{Z}_{\geq 0} \\ K^* < ||\bar{k}||_1}} \left( 2^{\langle \alpha, k \rangle} ||p_k||_{L^p} \right)^q \right)$$

$$\times \left( \sum_{\substack{k \in \mathbb{Z}_{\geq 0} \\ K < ||k||_1 \leq K^*}} \left( 2^{-(\underline{\alpha} - \delta - \delta\nu)(||k||_1 - K)} \right)^{\frac{qr}{q-r}} + \sum_{\substack{k \in \mathbb{Z}_{\geq 0} \\ K^* < ||\bar{k}||_1}} \left( 2^{-(\underline{\alpha} - \delta)(||k||_1 - K^*)} \right)^{\frac{qr}{q-r}} \right)^{\frac{q-r}{q}}$$

$$\lesssim 2^{-\underline{\alpha} K r} ||f||_{MB_{p,q}^\alpha}^r D_{K,d}^{r(\frac{1}{r} - \frac{1}{q})}.$$

**(2-3) the case of $r = \infty$**

We can execute the same the analysis as in the case of $q > r$. Since $\frac{1}{q} + \frac{1}{\frac{q}{q-1}} = 1$, then it follows by applying Hölder's inequality that

$$
\begin{aligned}
&||f - R_K(f)||_{L^\infty} \\
&\lesssim \sum_{\substack{k \in \mathbb{Z}_{\geq 0} \\ K < ||k||_1 \leq K^*}} \left( 2^{\delta ||k||_1} n_k^{-\delta} ||p_k||_{L^p} \right) + \sum_{\substack{k \in \mathbb{Z}_{\geq 0} \\ K^* < ||\bar{k}||_1}} \left( 2^{\delta ||k||_1} ||p_k||_{L^p} \right) \\
&\lesssim 2^{-\underline{\alpha} K} \Bigg( \sum_{\substack{k \in \mathbb{Z}_{\geq 0} \\ K < ||k||_1 \leq K^*}} \left( 2^{-(\underline{\alpha} - \delta - \delta \nu)(||k||_1 - K)} 2^{\langle \alpha, k \rangle} ||p_k||_{L^p} \right) \\
&\quad + \sum_{\substack{k \in \mathbb{Z}_{\geq 0} \\ K^* < ||\bar{k}||_1}} \left( 2^{-(\underline{\alpha} - \delta)(||k||_1 - K^*)} 2^{\langle \alpha, k \rangle} ||p_k||_{L^p} \right) \Bigg) \\
&\leq 2^{-\underline{\alpha} K} \Bigg( \sum_{\substack{k \in \mathbb{Z}_{\geq 0} \\ K < ||k||_1 \leq K^*}} \left( 2^{\langle \alpha, k \rangle} ||p_k||_{L^p} \right)^q + \sum_{\substack{k \in \mathbb{Z}_{\geq 0} \\ K^* < ||\bar{k}||_1}} \left( 2^{\langle \alpha, k \rangle} ||p_k||_{L^p} \right)^q \Bigg) \\
&\quad \times \Bigg( \sum_{\substack{k \in \mathbb{Z}_{\geq 0} \\ K < ||k||_1 \leq K^*}} (2^{-(\underline{\alpha} - \delta - \delta \nu)(||k||_1 - K)})^{\frac{q}{q-1}} + \sum_{\substack{k \in \mathbb{Z}_{\geq 0} \\ K^* < ||\bar{k}||_1}} \left( 2^{-(\underline{\alpha} - \delta)(||k||_1 - K^*)} \right)^{\frac{q}{q-1}} \Bigg)^{\frac{q-1}{q}} \\
&\lesssim 2^{-\underline{\alpha} K} ||f||_{MB_{p,q}^\alpha} D_{K,d}^{1 - \frac{1}{q}}.
\end{aligned}
$$

**Estimation of $\sharp E(K)$, $k_{\max}$ and $c_{\max}$:** First, we estimate the cardinality of $E(K)$. It follows from easy calculations that

$$
\begin{aligned}
\sharp E(K) &= \sum_{\kappa=0}^{K} 2^\kappa \binom{\kappa + d - 1}{d - 1} + \sum_{k: K < ||k||_1 \leq K^*} n_k \\
&\leq 2^{K+1} \binom{K + d - 1}{d - 1} + \sum_{K < \kappa \leq K^*} 2^{K - \nu(\kappa - K)} \binom{\kappa + d - 1}{d - 1} \\
&\leq 2^{K+1} D_{K,d} + 2^K (1 - 2^{-\nu})^{-1} D_{K^*,d} \\
&\leq \left( 2 + (1 - 2^{-\nu})^{-1} \right) 2^K D_{K^*,d}.
\end{aligned}
$$

Next, it clearly follows that $k_{\max} \leq ||k||_1 \leq K^*$.

Finally, we estimate $c_{\max}$. Since the inequality below holds

$$
2^{\left( \underline{\alpha} - \frac{1}{p} \right) ||k||_1} |c_{k,j}| \leq 2^{\langle \alpha, k \rangle - \frac{||k||_1}{p}} |c_{k,j}| \lesssim ||f||_{MB_{p,q}^\alpha},
$$

it follows that

$$
c_{\max} = \max_{\substack{1 \leq i \leq d \\ k \in E_K}} c_{k,j} \lesssim 2^{\left( \frac{1}{p} - \underline{\alpha} \right) ||k||_1} ||f||_{MB_{p,q}^\alpha} \leq 2^{K^* \left( \frac{1}{p} - \underline{\alpha} \right)_+} ||f||_{MB_{p,q}^\alpha}.
$$

This completes the proof. $\qquad \square$

**Theorem 7** (Cardinal B-spline approximation for anisotropic Besov spaces). *Suppose that $p, q, r \in \mathbb{R}_{>0} \cup \{\infty\}, \alpha \in (\mathbb{R}^d)^l$. Let $\delta := \left( \frac{1}{p} - \frac{1}{r} \right)_+$ (note that $\delta > 0$ is equivalent to $p < r$) and assume that $m \in \mathbb{Z}_{>0}$, and $\delta < \tilde{\alpha}, \bar{\alpha} < \min(m, m - 1 + \frac{1}{p})$.*

*Then, for any $f \in AB_{p,q}^\alpha$ and $K \in \mathbb{Z}_{\geq 1}$, there exist constants*

$$
\nu := \frac{\tilde{\alpha} - \delta}{2\delta}, \bar{N} := \left\lceil 2^{||K||_\alpha} \right\rceil, n_k := \left\lceil 2^{||K||_\alpha - \nu(||k||_\alpha - ||K||_\alpha)} \right\rceil, K^* := \left\lceil K \left( 1 + \nu^{-1} \right) \right\rceil,
$$

$$S(k) \subseteq J_m^d(k) \ such \ that \ \sharp(S(k)) = \left\lceil 2^{K - \nu(\|k\|_1 - K^*)} \right\rceil,$$

*where* $\|k\|_{\underline{\alpha}} := \sum_{i=1}^d \left\lfloor \frac{k\alpha}{\alpha i} \right\rfloor$, *and*

$$R_K(f) := \sum_{k=0}^{K} \sum_{j \in J^d(k)} c_{k,j} M_{k,j}^d(x) + \sum_{k=K+1}^{K^*} \sum_{j \in S(k)} c_{k,j} M_{k,j}^d(x)$$

*such that*

$$\|f - R_K(f)\|_r \lesssim \tilde{N}^{-\alpha} \|f\|_{AB_{p,q}^\alpha},$$

$$\sharp E(K) := \{(k,j) \in \mathbb{Z}_{\geq 0}^d \times \mathbb{Z}_{\geq 0}^d | c_{k,j} \neq 0\} \leq \left(2 + (1 - 2^{-\nu})^{-1}\right) \tilde{N},$$

$$k_{\max} := \max_{\substack{1 \leq i \leq d \\ k \in \bar{E}_K}} k_i \leq K^*, c_{\max} := \max_{\substack{1 \leq i \leq d \\ k \in \bar{E}_K}} c_{k,j} \lesssim 2^{K^* \left(\frac{d\bar{\alpha}}{p} - \underline{\alpha}\right)_+} \|f\|_{AB_{p,q}^\alpha}.$$

*Moreover, it follows that* $2^K \leq \tilde{N}$.

*Proof.* **For the existence of** $R_K(f)$**:** See the proof in Suzuki & Nitanda (2021).

**Estimation of** $\sharp E(K)$**,** $k_{\max}$ **and** $c_{\max}$**:** First, we estimate the cardinality of $E(K)$. It follows from easy calculations that

$$\begin{aligned}
\sharp E(K) &= \sum_{k=0}^{K} 2^k + \sum_{k: K < \|k\|_1 \leq K^*} n_k \\
&\leq 2^{K+1} + \sum_{K < \kappa \leq K^*} 2^{K - \nu(\kappa - K)} \\
&\leq 2^{K+1} + 2^K (1 - 2^{-\nu})^{-1} \\
&\leq \left(2 + (1 - 2^{-\nu})^{-1}\right) \bar{N}.
\end{aligned}$$

Next, since it clearly follows that $k_{\max} \leq K^*$, we estimate $c_{\max}$. Since the inequality below holds

$$2^{k\left(\underline{\alpha} - \frac{\sum_{i=1}^d \lfloor k\alpha_i \rfloor}{kp}\right)} |c_{k,j}| \lesssim \|f\|_{AB_{p,q}^\alpha},$$

it follows that

$$c_{\max} = \max_{\substack{1 \leq i \leq d \\ k \in \bar{E}_K}} c_{k,j} \lesssim 2^{k\left(\frac{\sum_{i=1}^d \lfloor k\alpha_i \rfloor}{kp} - \underline{\alpha}\right)_+} \|f\|_{AB_{p,q}^\alpha} \leq 2^{K^* \left(\frac{d\bar{\alpha}}{p} - \underline{\alpha}\right)_+} \|f\|_{AB_{p,q}^\alpha}.$$

Finally, it clearly follows that $2^K \leq 2^{\|K\|_{\underline{\alpha}/\alpha}} \leq N$.

This completes the proof. $\qquad \square$

By using the results of Theorem 6 and Theorem 7, we can prove Theorem 1, which shows the approximation ability of Transformer networks for the target function in a mixed smooth Besov space or an anisotropic Besov space.

*Proof of Theorem 1.* Let

$$g := \sum_{n=1}^{N} c_n M_{k_n, j_n}^{dl}(x),$$

and we define

$$k_{\max} := \max_{1 \leq n \leq N} k_n, c_{\max} := \max_{1 \leq n \leq N} c_n.$$

Fix $\varepsilon \in \mathbb{R}_{>0}$. Now, we construct an approximate Transformer network $T[g]$ of $g$.

It follows from Lemma 3 that, for any $\epsilon, R \in \mathbb{R}_{>0}$, there exist constants

$$\bar{L}_{\epsilon,R} := \left\lceil \log_2 \left( \frac{R}{\epsilon} \right) \right\rceil, \bar{W}_{\epsilon,R} := 3, \bar{S}_{\epsilon,R} := 5\bar{L}_{\epsilon,R}, \bar{B}_{\epsilon,R} := 1,$$

and a neural network $\bar{M}_{\epsilon,R} \in \Phi_2(\bar{L}_{\epsilon,R}, \bar{W}_{\epsilon,R}, \bar{S}_{\epsilon,R}, \bar{B}_{\epsilon,R})$, such that

$$\sup_{x,y \in [0,R]} \left| xy - \bar{M}_{\epsilon,R}(x) \right| \leq \epsilon.$$

**Positional encoding and B-spline coefficients:** Let $\varepsilon_1 := \frac{\varepsilon}{4dlc_{\max}}$,

$$P_E := I, K := \begin{pmatrix} k_1 \\ \vdots \\ k_N \end{pmatrix}, J := \begin{pmatrix} j_1 \\ \vdots \\ j_N \end{pmatrix},$$

$$M_{1,1} := \begin{pmatrix} \begin{pmatrix} I_d \\ O \\ O \end{pmatrix} & \begin{pmatrix} O_d \\ K \\ I \end{pmatrix} \end{pmatrix}, M_{1,2} := \begin{pmatrix} I_{Nd} & -J \\ O & I \end{pmatrix}.$$

Let $M_1 := M_{1,2} \circ \bar{M}_{\varepsilon_1, 2^{k_{\max}}} \circ M_{1,1}, \tilde{M}_1 := \Pi(M_1) \circ \Pi(M) \circ \mathbf{Concat}[P_E]$. Then, it follows that, for any integer $j(1 \leq j \leq l)$, For $k, j \in \mathbb{Z}_{\geq 0}^d, x \in \mathbb{R}^d$, it follows that

$$\left\| \begin{pmatrix} 2^{k_1} x - j_1 \\ \vdots \\ 2^{k_h} x - j_h \end{pmatrix} - \tilde{M}_1(x) \right\|_{L^\infty} \leq \frac{\varepsilon}{4dlc_{\max}}$$

where

$$2^k x - j := \begin{pmatrix} 2^{k_1} x_1 - j_1 \\ \vdots \\ 2^{k_d} x_d - j_d \end{pmatrix}.$$

**Token-wise B-splines:** Let $y_1, \ldots, y_H \in \mathbb{R}^d, \varepsilon_2 := \frac{\varepsilon}{4lc_{\max}}$.

It follows from Lemma 5 that there exist constants

$$L_{2,\varepsilon_2} := 3 + 2 \left\lceil \log_2 \left( \frac{3^{d \vee m}}{\epsilon c_{(d,m)}} \right) + 5 \right\rceil \left\lceil \log_2(d \vee m) \right\rceil,$$

$$W_{2,\varepsilon_2} := 6dm(m+2) + 2d, S_{2,\varepsilon_2} := L_0 W_0^2, B_{2,\varepsilon_2} := 2(m+1)^m,$$

and a neural network $M_{2,1} \in \Phi_2(L_{2,\varepsilon_2}, W_{2,\varepsilon_2}, S_{2,\varepsilon_2}, B_{2,\varepsilon_2})$ such that

$$\|M_{0,0}^d - M_{2,1}\|_{L^\infty(\mathbb{R}^D)} \leq \varepsilon_2.$$

Then we define

$$M_2 \left( \begin{pmatrix} y_1 \\ \vdots \\ y_H \\ e \end{pmatrix} \right) := \begin{pmatrix} M_{2,1}(y_1) \\ \vdots \\ M_{2,1}(y_H) \\ e \end{pmatrix},$$

and $\tilde{M}_2 := \Pi(M_2) \circ \tilde{M}_1$. Since it holds from Lemma 1 that, for any $r \in \mathbb{R}$, $\mathcal{N}_m(x)$ is 1-Lipschitz and $\|\mathcal{N}_m(x)\|_{L^\infty} \leq 1$, it follows that

$$\left\| \begin{pmatrix} M_{k_1,j_1}^d(x_1) & \cdots & M_{(k_1)_l,(j_1)_l}^d(x_l) \\ \vdots & \ddots & \vdots \\ M_{(k_h)_1,(j_h)_1}^d(x_1) & \cdots & M_{(k_h)_l,(j_h)_l}^d(x_l) \end{pmatrix} - \tilde{M}_2(x) \right\|_{L^\infty(([0,1]^d)^l)}$$

$$\leq \sup_{1 \leq h \leq H} \left\| M_{0,0}^d(2^{k_h} x - j_h) - M_{0,0}^d(\tilde{M}_{1,h}(x)) \right\|_{L^\infty(([0,1]^d)^l)}$$

$$+ \sup_{1 \leq h \leq H} \left\| M_{0,0}^d(\tilde{M}_{1,h}(x)) - M_{b,\varepsilon_2}(\tilde{M}_{1,h}(x_i)) \right\|_{L^\infty(\mathbb{R}^d)}$$

$$\leq \frac{\varepsilon}{4lc_{\max}} + \frac{\varepsilon}{4lc_{\max}} \leq \frac{\varepsilon}{2lc_{\max}}.$$

**Multiplication between tokens:** Let $T_3 := \lceil \log_2 l \rceil$, $\varepsilon_3 := \frac{\varepsilon}{6T_3 c_{\max}}$, $\gamma := \log\left(\frac{l}{\varepsilon_3}\right)$, and $\delta := \frac{e^\gamma}{(l-1)+e^\gamma}$. Note that

$$1 \geq \delta = \frac{e^\gamma}{(l-1)+e^\gamma} \geq 1 - le^{-\gamma} = 1 - \varepsilon_3.$$

$$W_{3,1} := \left( \begin{pmatrix} -I & O \\ -I & O \\ & O \end{pmatrix} \begin{pmatrix} \mathbb{1} & \mathbb{0} & \cdots \\ \mathbb{0} & \mathbb{1} & \cdots \\ & I \end{pmatrix} \right), M_{3,1}(x) := x + W_{3,1}((x)_+)$$

If all the elements $x_{ij}$ of $x_i$ are $0 \leq x_{ij} \leq 1$, then it follows from easy calculations that

$$M_{3,1}\left( \left( \begin{pmatrix} x_1 \\ 0 \\ e_1 \end{pmatrix} \begin{pmatrix} x_2 \\ 0 \\ e_2 \end{pmatrix} \cdots \right) \right) := \left( \begin{pmatrix} \mathbb{1}-x_1 \\ -x_1 \\ e_1 \end{pmatrix} \begin{pmatrix} -x_2 \\ \mathbb{1}-x_2 \\ e_2 \end{pmatrix} \cdots \right),$$

$$(M_{3,1} \circ M_{3,1})\left( \left( \begin{pmatrix} x_1 \\ 0 \\ e_1 \end{pmatrix} \begin{pmatrix} x_2 \\ 0 \\ e_2 \end{pmatrix} \cdots \right) \right) := \left( \begin{pmatrix} x_1 \\ 0 \\ e_1 \end{pmatrix} \begin{pmatrix} 0 \\ x_2 \\ e_2 \end{pmatrix} \cdots \right).$$

Let $H := 1$, $M_K^{(1)} := \begin{pmatrix} O & \begin{pmatrix} B & B & \cdots \\ B & B & \cdots \\ \vdots & \vdots & \ddots \end{pmatrix} \end{pmatrix}$, $M_Q^{(1)} = (O \quad I)$, $M_V^{(1)} = \begin{pmatrix} I & O \\ O & O \end{pmatrix}$, $M_O^{(1)} := I$

where $B := \gamma \begin{pmatrix} 0 & 0 \\ 1 & 0 \end{pmatrix}$. Since

$$\Pi(M_V^{(1)})(x) = \begin{pmatrix} x_1 & x_2 & \cdots \\ 0 & 0 & \cdots \end{pmatrix}, \Pi(M_K^{(1)})(x) = \gamma \begin{pmatrix} B & O & \cdots \\ O & B & \cdots \\ \vdots & \vdots & \ddots \end{pmatrix}, \Pi(M_Q^{(1)})(x) = I,$$

then, it follows that

$$\mathbf{Attn}(\Pi(M_V^{(1)})(x), \Pi(M_K^{(1)})(x), \Pi(M_Q^{(1)})(x)) = \delta \begin{pmatrix} x_2 & \frac{1}{l}\sum_{i=1}^l x_i & x_4 & \frac{1}{l}\sum_{i=1}^l x_i & \cdots \\ 0 & 0 & 0 & 0 & \cdots \end{pmatrix} + R$$

where $R = \begin{pmatrix} \frac{1-\delta}{l-1}\left(\left(\sum_{i=1}^l x_i\right) - x_2\right) & 0 & \frac{1-\delta}{l-1}\left(\left(\sum_{i=1}^l x_i\right) - x_4\right) & 0 & \cdots \\ 0 & 0 & 0 & 0 & \cdots \end{pmatrix}$. Thus, it holds that

$$M_{3,2}\left( \left( \begin{pmatrix} x_1 \\ 0 \\ e_1 \end{pmatrix} \begin{pmatrix} 0 \\ x_2 \\ e_2 \end{pmatrix} \cdots \right) \right)$$

$$= \left( \begin{pmatrix} x_1 \\ 0 \\ e_1 \end{pmatrix} \begin{pmatrix} 0 \\ x_2 \\ e_2 \end{pmatrix} \cdots \right) + \Pi(M_O^{(1)})(\mathbf{Attn}(\Pi(M_V^{(1)})(x), \Pi(M_K^{(1)})(x), \Pi(M_Q^{(1)})(x)))$$

$$= \left( \begin{pmatrix} x_1 \\ 0 \\ e_1 \end{pmatrix} \begin{pmatrix} 0 \\ x_2 \\ e_2 \end{pmatrix} \cdots \right) + \delta \left( \begin{pmatrix} 0 \\ x_2 \\ 0 \end{pmatrix} \ast \begin{pmatrix} 0 \\ x_4 \\ 0 \end{pmatrix} \ast \cdots \right) + R$$

$$= \left( \begin{pmatrix} x_1 \\ \delta x_2 \\ 0 \end{pmatrix} \ast \begin{pmatrix} x_3 \\ \delta x_4 \\ 0 \end{pmatrix} \ast \cdots \right) + R$$

Let $M_{3,3} := \bar{M}_{\varepsilon_3,1}$ and $M_3^1 := \Pi(M_{3,3}) \circ M_{3,2} \circ \Pi(M_{3,1} \circ M_{3,1})$. We denote $m_{h,i}$ by the $i$-th element of $M_{0,0}^d(2^{k_h}x - j_h)$, $M_{2,h,i}$ by the $i$-th element of $\tilde{M}_2(x)$, and $M_{3,h,i}$ by the $i$-th element

of $\left( M_3^1 \circ \tilde{M}_2 \right)(x)$. Then, the following inequality holds:

$$\sup_{x \in [0,1]^{dl}} |m_{h,2i-1}(x)m_{h,2i}(x) - M_{3,h,2i-1}(x)|$$

$$\leq \sup_{1 \leq h \leq H} |m_{h,2i-1}(x)m_{h,2i}(x) - M_{2,h,2i-1}(x)m_{h,2i}(x)|$$

$$+ \sup_{1 \leq h \leq H} |M_{2,h,2i-1}(x)m_{h,2i}(x) - M_{2,h,2i-1}(x)M_{2,h,2i}(x)|$$

$$+ \sup_{1 \leq h \leq H} |M_{2,h,2i-1}(x)M_{2,h,2i}(x) - M_{2,h,2i-1}(x)\delta M_{2,h,2i}(x)|$$

$$+ \sup_{1 \leq h \leq H} |M_{2,h,2i-1}(x)\delta M_{2,h,2i}(x) - M_{3,h,2i-1}(x)|$$

$$\leq \sup_{1 \leq h \leq H} |m_{h,2i-1}(x) - M_{2,h,2i-1}(x)| + \sup_{1 \leq h \leq H} |m_{h,2i}(x) - M_{2,h,2i}(x)| + (1 - \delta) + (\varepsilon_3 + (1 - \delta))$$

$$\leq \frac{2}{l}\frac{\varepsilon}{2c_{\max}} + \frac{1}{T_3}\frac{\varepsilon}{2c_{\max}}.$$

Next, as well as $M_3^1$, we can sequentially construct $M_3^2, \ldots, M_3^{T_3}$ (we can execute these construction by replacing $B$ by $\begin{pmatrix} 0 & 0 & 0 & 0 \\ 0 & 0 & 0 & 0 \\ 1 & 0 & 0 & 0 \\ 0 & 0 & 0 & 0 \end{pmatrix}, \ldots$), and define $M_3 := M_3^{T_3} \circ \cdots \circ M_3^1$ and $\tilde{M}_3 := M_3 \circ \tilde{M}_2$

such that

$$\left\| M_{k,j}^{dl}(x) - \tilde{M}_3(x) \right\|_{L^\infty(([0,1]^d)^l)} = \left\| \prod_{i=1}^{l} m_{h,i}(x) - \tilde{M}_3(x) \right\|_{L^\infty(([0,1]^d)^l)} \leq \frac{\varepsilon}{c_{\max}}.$$

**Linear combination of B-splines:** Let $M' := \begin{pmatrix} c_1 & \cdots & c_{N'} \end{pmatrix}$. Finally, we define an approximate Transformer network $T[g]$ of $g$.

$$T[g] := \mathbf{Head} \circ \Pi(M') \circ \tilde{M}_3.$$

Then, it follows from the estimations above that

$$\sup_{x \in [0,1]^{dl}} |g(x) - T[g](x)| \leq \varepsilon.$$

**Estimation of the bounds of hyper-parameters:** Let $f \in \begin{cases} MU_{p,q}^\alpha \\ AU_{p,q}^\alpha \end{cases}$. Then, it follows from Theorem 6 and Theorem 7 that there exist constants

$$D := \begin{cases} D_{K,dl}, \\ 1, \end{cases} \quad D^* := \begin{cases} D_{K^*,dl}, \\ 1, \end{cases} \quad \hat{N} := \begin{cases} 2^K, \\ \tilde{N}, \end{cases} \quad \hat{\alpha} := \begin{cases} \alpha, \\ \tilde{\alpha}, \end{cases}$$

$$N^* := (2 + (1 - 2^{-\nu})^{-1})\hat{N}D^*, \gamma := \begin{cases} \left( \frac{1}{p} - \underline{\alpha} \right)_+, \\ \left( \frac{dl\bar{\alpha}}{p} - \underline{\alpha} \right)_+, \end{cases}$$

$$\eta := \begin{cases} \left( \frac{1}{\min(r,1)} - \frac{1}{q} \right)_+ & (r \leq p), \\ \left( \frac{1}{r} - \frac{1}{q} \right)_+ & (p < r \text{ and } r < \infty), \\ \left( 1 - \frac{1}{q} \right)_+ & (p < r \text{ and } r = \infty), \end{cases}$$

and an approximation function

$$R_K(f)(x) := \sum_{n=1}^{N} c_n M_{k_n,j_n}^{dl}(x).$$

such that

$$\|f - R_K(f)\|_r \lesssim \hat{N}^{-\hat{\alpha}} D^\eta,$$

$$N \leq N^*, k_{\max} := \max_{1 \leq n \leq N} k_n \leq K^*, c_{\max} := \max_{1 \leq n \leq N} c_n \lesssim 2^{K^*\gamma}, 2^K \leq \hat{N}.$$

Let $\varepsilon := \hat{N}^{-\hat{\alpha}}$. Then, from above estimations, it immediately follows that

$$||f - T[R_K(f)]||_r \lesssim ||f - R_K(f)||_r + ||R_K(f) - T[R_K(f)]||_r \lesssim \hat{N}^{-\hat{\alpha}}D^\eta.$$

Second, it follows from that there exist absolute constant $C \in \mathbb{R}_{>0}$. Third, we define constants which define hyper-parameters as follows:

$$\epsilon := \hat{N}^{-(\hat{\alpha}+(1+\nu^{-1})\gamma)}, T_0 := \lceil \log_2 l \rceil, L_1 = C \left\lceil \log\left(\frac{4dl}{\epsilon}\right)\right\rceil,$$

$$L_2 = 3 + 2\left\lceil \log_2\left(\frac{4l \cdot 3^{d \vee m}}{\epsilon c_{(d,m)}}\right) + 5\right\rceil \lceil \log_2(d \vee m)\rceil, L_3 = C\left\lceil \log\left(\frac{6T_0}{\epsilon}\right)\right\rceil,$$

$$W_0 := 6dm(m+2) + 2d, \zeta := \begin{cases} 1, \\ \bar{\alpha}. \end{cases}$$

Note that $\epsilon \lesssim \frac{\varepsilon}{c_{\max}}$.

Finally, we estimate hyper-parameters of the Transformer network. Because of the above estimations, it immediately follows that

$$
\begin{aligned}
T &= T_0, \\
E &= l, \\
H &= 1, \\
L &= L_1 + L_2 + T(L_3 + 2) + 1 \\
W &= W_0 N^* + E \\
S &\leq C(N^*l + L_1) + L_2 W_0^2 N^* + CT((N^* + E) + L_3) + N^*, \\
B &\lesssim \hat{N}^{(1+\nu^{-1})(\gamma \vee \zeta)}.
\end{aligned}
$$

This completes the proof. $\qquad\square$

## D    PROOF OF THE STATEMENTS OF SECTION 4

To prove Theorem 3, we define a covering number with respect to a given metric space.

**Definition 13** (Covering number and packing number). *Let $\epsilon \in \mathbb{R}_{>0}$, $(V, d)$ be a metric space, and $\mathcal{F}$ be a subset of $V$. Then, we define a covering number $\mathcal{N}(\epsilon, \mathcal{F}, (V, d))$ of $\mathcal{F}$ with respect to $(V, d))$ as follows:*

$$\mathcal{N}(\epsilon, \mathcal{F}, (V, d)) := \min\left\{ N \middle| N = \sharp(K), K \subseteq \mathcal{F}, \mathcal{F} \subseteq \bigcup_{f_i \in K} \{f \in V | d(f, f_i) \leq \epsilon\} \right\}.$$

Before proving Theorem 3, we have to estimate the covering number of the set of transformer networks $\mathbf{TN}(L, T, E, W, H, S, B)$ with respect to $L^\infty$.

**Lemma 6** (Covering number estimation of the set of Transformer networks). *The covering number of $\mathbf{TN}(L, T, E, W, H, S, B)$ can be bounded by*

$$\log \mathcal{N}(\delta, \mathbf{TN}(L, T, E, W, H, S, B), ||\cdot||_{L^\infty})$$
$$\leq S \log\left(4\delta^{-1}(L + T + 1)(W + 1)^{2L+2T}(B + 1)^{L+2T+1}H^T\right).$$

*Proof.* First, we define
$$\mathcal{F}[M, b](x) := x + M \cdot (x)_+ + b.$$

Then, the following estimations hold:

$$\|\mathcal{F}[M, b](x)\|_{L^\infty}$$
$$\leq \|x\|_\infty + \max_j \|M_{j,:}\|_1 \|x\|_\infty + |b| \leq (WB + 1) \|x\|_\infty + B$$
$$\leq (W + 1)(B + 1)(\|x\|_\infty \vee 1),$$
$$\|\mathcal{F}[M, b](x_1) - \mathcal{F}[M, b](x_2)\|_{L^\infty}$$
$$\leq \|x_1 - x_2\|_\infty + \max_j \|M_{j,:}\|_1 \|(x_1)_+ - (x_2)_+\|_\infty \leq \|x_1 - x_2\|_\infty + WB \|x_1 - x_2\|_\infty$$
$$\leq (WB + 1) \|x_1 - x_2\|_\infty,$$

and, if $\delta' := \|M - M'\|_\infty \vee \|b - b'\|_\infty$, then it follows that

$$\|\mathcal{F}[M, b](x) - \mathcal{F}[M', b'](x)\|_{L^\infty} \leq \delta (W \|x\|_\infty + 1).$$

Next, we define

$$\mathcal{A}[M_*^h](x) := x + \sum_{h=1}^{H} \Pi(M_O^h)(\mathbf{Attn}(\Pi(M_V^h)(x), \Pi(M_K^h)(x), \Pi(M_Q^h)(x))).$$

Then, the following estimations holds

$$\left\|\mathcal{A}[M_*^h](x)\right\|_{L^\infty}$$
$$\leq \|x\|_\infty + H \max_j \left\|(M_O^h)_{j,:}\right\|_1 \max_j \left\|(M_V^h)_{j,:}\right\|_1 \|x\|_\infty$$
$$\leq (HW^2B^2 + 1) \|x\|_\infty,$$
$$\left\|\mathcal{A}[M_*^h](x_1) - \mathcal{A}[M_*^h](x_2)\right\|_{L^\infty}$$
$$\leq \|x_1 - x_2\|_\infty + H \max_j \left\|(M_O^h)_{j,:}\right\|_1 \max_j \left\|(M_V^h)_{j,:}\right\|_1 \|x_1 - x_2\|_\infty.$$
$$\leq \|x_1 - x_2\|_\infty + HW^2B^2 \|x_1 - x_2\|_\infty$$
$$\leq (HW^2B^2 + 1) \|x_1 - x_2\|_\infty,$$

and, if $\delta' := \max_{1 \leq h \leq H, * \in \{O, V, K, Q\}} \left\|M_*^h - (M')_*^h\right\|_\infty$, then it follows that

$$\left\|\mathcal{A}[M_*^h](x) - \mathcal{A}[(M')_*^h](x)\right\|_{L^\infty} \leq 2\delta HW^2B \|x\|_\infty.$$

Next, we define

$$\mathcal{F}_k(x) := (\mathcal{L}_k \circ \cdots \circ \mathcal{L}_0)(x), \mathcal{F}'_k(x) := (\mathcal{L}'_k \circ \cdots \circ \mathcal{L}'_0)(x),$$
$$\mathcal{B}_k(x) := (\mathcal{L}_{L+T} \circ \cdots \circ \mathcal{L}_k)(x), \mathcal{B}'_k(x) := (\mathcal{L}'_{L+T} \circ \cdots \circ \mathcal{L}'_k)(x)$$

where, for $k = 0$, $\mathcal{L}_k(x) := \mathbf{Concat}[P_E](x)$ and $\mathcal{L}'_k(x) := \mathbf{Concat}[P'_E](x)$, and, for $k \geq 1$, either one of the following cases is true:

1. $\mathcal{L}_k(x) := \Pi(\mathcal{F}[M_k, b_k])(x)$ and $\mathcal{L}'_k(x) := \Pi(\mathcal{F}[M'_k, b'_k])(x)$,

2. $\mathcal{L}_k(x) := \mathcal{A}[(M_k)_*^h](x)$ and $\mathcal{L}'_k(x) := \mathcal{A}[(M'_k)_*^h](x)$.

Moreover, for $k \geq 1$, we denote

$$f_k := \sharp\{i \in \mathbb{Z} \| 1 \leq i \leq k, \mathcal{L}_i(x) := \mathcal{A}[(M_i)_*^h](x)\},$$
$$l_k := \sharp\{i \in \mathbb{Z} \| i = k, \mathcal{L}_i(x) := \mathcal{A}[(M_i)_*^h](x)\},$$
$$b_k := \sharp\{i \in \mathbb{Z} \| k \leq i \leq L + T, \mathcal{L}_i(x) := \mathcal{A}[(M_i)_*^h](x)\}.$$

If $f = \mathcal{F}_{L+T}(x), f' = \mathcal{F}'_{L+T}(x)(f, f' \in \mathbf{TN}(L, T, E, W, H, S, B))$, and

$$\delta' := \max_{1 \leq k \leq L+T} \left\{ \|P_E - P'_E\|_\infty \vee \|M_k - M'_k\|_\infty \vee \|b_k - b'_k\|_\infty \right.$$
$$\left. \vee \max_{1 \leq h \leq H, * \in \{O, V, K, Q\}} \left\|(M_k)_*^h - (M'_k)_*^h\right\|_\infty \right\},$$

then it follows that

$$\|f - f'\|_{L^\infty}$$

$$\leq \sum_{k=0}^{L+T} \left\| \mathcal{B}'_{k+1} \circ \mathcal{L}_k \circ \mathcal{F}_{k-1} - \mathcal{B}'_{k+1} \circ \mathcal{L}'_k \circ \mathcal{F}_{k-1} \right\|_{L^\infty}$$

$$\leq \sum_{k=0}^{L+T} (WB+1)^{L+T-k-b_{k+1}} (HW^2B^2+1)^{b_{k+1}} \left\| \mathcal{L}_k \circ \mathcal{F}_{k-1} - \mathcal{L}'_k \circ \mathcal{F}_{k-1} \right\|_{L^\infty}$$

$$\leq (WB+1)^L (HW^2B^2+1)^T \delta'$$

$$+ \sum_{k=1}^{L+T} (WB+1)^{L+T-k-b_{k+1}} (HW^2B^2+1)^{b_{k+1}} \delta' (W+1)^{l_k} (2HW^2B)^{l_k} \left( \|\mathcal{F}_{k-1}\|_{L^\infty} \vee 1 \right)$$

$$\leq (WB+1)^L (HW^2B^2+1)^T \delta'$$

$$+ \sum_{k=1}^{L+T} (WB+1)^{L+T-k-b_{k+1}} (HW^2B^2+1)^{b_{k+1}} \delta' (W+1)^{l_k} (2HW^2B)^{l_k}$$

$$\cdot (W+1)^{k-1-f_{k-1}} (B+1)^{k-1-f_{k-1}} (HW^2B^2+1)^{f_{k-1}} \left( \|\mathcal{L}_0\|_\infty \vee 1 \right)$$

$$\leq \sum_{k=0}^{L+T} 2\delta' H^T (W+1)^{L+2T} (B+1)^{L+2T-1} (B \vee 1)$$

$$\leq 2\delta' (L+T+1) H^T (W+1)^{L+2T} (B+1)^{L+2T}$$

Thus, for a fixed sparsity pattern (the locations of non-zero parameters), the covering number is bounded by

$$\left( 2B \cdot \frac{2(L+T+1)H^T (W+1)^{L+2T} (B+1)^{L+2T}}{\delta} \right)^S.$$

Thus, the covering number of the whole space $\mathbf{TN}(L, T, E, W, H, S, B)$ is bounded as

$$\binom{(W+1)^L}{S} \left( 2B \cdot \frac{2(L+T+1)H^T (W+1)^{L+2T} (B+1)^{L+2T}}{\delta} \right)^S$$

$$\leq \left( 4\delta^{-1} H^T (L+T+1)(W+1)^{2L+2T} (B+1)^{L+2T+1} \right)^S.$$

This completes the proof. $\qquad\square$

Next, to prove Theorem 3, we need the following result which connects the approximation theory to generalization error analysis.

**Proposition 1** (Schmidt-Hieber (2017)). *Let $\mathcal{F}$ be a set of functions, let $\hat{f}$ be any estimator in $\mathcal{F}$. Define*

$$\Delta_n := \mathbb{E}_{D_n} \left( \frac{1}{n} \sum_{i=1}^n (y_i - \hat{f}(x_i))^2 - \inf_{f \in \mathcal{F}} \left( \frac{1}{n} \sum_{i=1}^n (y_i - \hat{f}(x_i))^2 \right) \right).$$

*Assume that $\|f^\circ\| \leq F$ and all $f \in \mathcal{F}$ satisfies $\|f\|_{L^\infty} \leq F$ for some $F \geq 1$. If $0 < \delta < 1$ satisfies $\mathcal{N}(\delta, \mathcal{F}, \|\cdot\|_{L^\infty})$, then there exists a universal constant $C$ such that*

$$\mathbb{E}_{D_n} \left( \|f^\circ - \hat{f}\|_{L^2(P_X)} \right)$$

$$\leq C(1+\epsilon)^2 \left( \inf_{f \in \mathcal{F}} \|f - f^\circ\|_{L^2(P_X)}^2 + F^2 \frac{\log \mathcal{N}(\delta, \mathcal{F}, \|\cdot\|_{L^\infty}) - \log \delta}{n\epsilon} + \delta F^2 + \Delta_n \right)$$

*for any $\epsilon \in (0, 1]$.*

*Proof.* See the proof in Schmidt-Hieber (2017) and Suzuki (2019). $\qquad\square$

Finally, based on the definition and the results above, we can prove Theorem 3.

*Proof of Theorem Theorem 3 and Theorem 4.* Note that the estimations below holds:

$$D \lesssim D^*, L \lesssim \log \hat{N}, T \lesssim 1, W \lesssim \hat{N}D, H \lesssim 1, S \lesssim \hat{N}D \log \hat{N}, B \lesssim \hat{N}^{(1+\nu^{-1})(\gamma\vee\zeta)}.$$

Lemma 6 gives an upper bound of the covering number as

$$\log \mathcal{N}(\delta, \Phi(L, T, E, W, H, S, B), ||\cdot||_{L^\infty})$$
$$\leq S \log \left(4\delta^{-1}(L+T+1)(W+1)^{2L+2T}(B+1)^{L+2T+1}H^T\right)$$
$$\lesssim \hat{N}D \log \hat{N} \left((\log \hat{N})^2 + \log\left(\delta^{-1}\right)\right).$$

Next, it follows from Theorem 1 that

$$||f - R_K(f)||_{L^2} \lesssim \hat{N}^{-\hat{\alpha}}D^\eta$$

where $\eta := \eta_{p,q,r}$. Since $P_X$ has a density function $0 \leq p(x) < R$ on $([0,1]^d)^l$, then it holds that

$$||f - R_K(f)||_{L^2(P_X)} \lesssim ||f - R_K(f)||_{L^2}$$

for any $f : ([0,1]^d)^l \to \mathbb{R}$, and by applying Proposition 1 with $\delta = \frac{1}{n}$, it follows that

$$\mathbb{E}_{D_n}\left(||f^\circ - \hat{f}||_{L^2(P_X)}\right) \lesssim \hat{N}^{-2\hat{\alpha}}D^{2\eta} + \frac{\hat{N}D \log \hat{N}\left((\log \hat{N})^2 + \log n\right)}{n} + \frac{1}{n}.$$

**The case of mixed smooth Besov spaces:** Note that $\hat{N} = 2^K, D = D_{k,dl}$. Since

$$D_{k,dl} := \left(1 + \frac{dl-1}{k}\right)^k \left(1 + \frac{k}{dl-1}\right)^{dl-1} \lesssim K^{dl-1}$$

then we obtain the following upper bound estimation:

$$\mathbb{E}_{D_n}\left(||f^\circ - \hat{f}||_{L^2(P_X)}\right) \lesssim 2^{-2\underline{\alpha}K}K^{2\eta(dl-1)} + \frac{K^{dl}2^K(K^2 + \log n)}{n}.$$

Then, the right hand side is minimized by $K = \left\lceil \frac{1}{1+2\underline{\alpha}}\log_2 n + \frac{(2\eta-1)(dl-1)+3}{1+2\underline{\alpha}}\log_2 \log n \right\rceil$ up to $\log \log n$-order. Then we obtain the following result:

$$\lesssim n^{-\frac{2\underline{\alpha}}{2\underline{\alpha}+1}}(\log n)^{\frac{2(dl-1)(\eta+\underline{\alpha})+6\underline{\alpha}}{1+2\underline{\alpha}}}.$$

**The case of anisotropic Besov spaces:** Note that $\hat{N} = \tilde{N}, D = 1$. The right hand side is minimized by $\tilde{N} = \left\lceil n^{\frac{1}{1+2\hat{\alpha}}}(\log n)^{-\frac{3}{1+2\hat{\alpha}}} \right\rceil$ up to $\log \log(n)$-order. Then we obtain the following result:

$$\lesssim n^{-\frac{2\hat{\alpha}}{2\hat{\alpha}+1}}(\log n)^{\frac{6\hat{\alpha}}{1+2\hat{\alpha}}}.$$

This completes the proof. $\qquad\square$

## E  PROOF OF THE STATEMENTS OF SECTION 5

*Proof of Theorem 5.* In the proof of Theorem 6, we can easily check that how to construct $R_K(f)$ is independent of the smoothness parameter $\alpha$. And, since $\underline{\alpha} = s$, as in the proof of Theorem 6, we obtain the following estimation

$$||f - R_K(f)||_r \lesssim 2^{-sK}D_{K,d}^{\eta_{p,q,r}}||f||_{VB_{p,q}^{\alpha,\pi}}.$$

such that

$$R_K(f) := \sum_{(k,j)\in E(K)} c_{k,j}M_{k,j}^d(x),$$

$$\sharp E(K) := \{(k,j) \in \mathbb{Z}_{\geq 0}^d \times \mathbb{Z}_{\geq 0}^d | c_{k,j} \neq 0\} \leq \left(2 + (1-2^{-\nu})^{-1}\right)2^K D_{K^*,d}.$$

From now on, we consider the case in which $\pi = \{\Omega\}$. For a general $\pi$, we can adapt the same proof strategy. Let $t_0 := \left\lceil \frac{\log(2K + \frac{1}{p})}{\log s} \right\rceil$ ($\iff s^{t_0} - \frac{1}{p} \geq 2K$) and

$$\tilde{R}_K(f) := \sum_{\substack{(k,j) \in E(K) \\ k_{\cdot,i'} \neq 0, \sigma(i') < t_0}} c_{k,j} M_{k,j}^d(x).$$

If $k_{\cdot,i'} \neq 0, \sigma(i') \geq t_0$, since $s \geq \frac{1}{p}$, the following inequality holds:

$$2^{2K}|c_{k,j}| \leq 2^{\left(s^{t_0} - \frac{1}{p}\right)}|c_{k,j}| \leq 2^{\langle \alpha, k \rangle - \frac{\|k\|_1}{p}}|c_{k,j}| \lesssim \|f\|_{MB_{p,q}^\alpha}.$$

Thus, since $r \geq 1$, it follows that

$$\left\| R_K(f) - \tilde{R}_K(f) \right\|_r \leq \sum_{\substack{(k,j) \in E(K) \\ k_{\cdot,i'} \neq 0, \sigma(i') \geq t_0}} |c_{k,j}| \left\| M_{k,j}^d \right\|_r \leq \sharp E(K) 2^{-2K} \|f\|_{VB_{p,q}^{\alpha,\pi}}$$

$$\leq \left(2 + (1 - 2^{-\nu})^{-1}\right) 2^K D_{K^*,d} 2^{-2K} \|f\|_{VB_{p,q}^{\alpha,\pi}} \lesssim 2^{-K} D_{K^*,d} \|f\|_{VB_{p,q}^{\alpha,\pi}}.$$

From estimations above, it follows that

$$\left\| f - \tilde{R}_K(f) \right\|_r \lesssim 2^{-K} D_{K,d}^{\eta_{p,q,r} \vee 1} \|f\|_{VB_{p,q}^{\alpha,\pi}}.$$

Let

$$\overline{\mathcal{N}_m}(x) := \sum_{i=1}^{2^K(m+1)} \mathbb{1}_{[2^{-K}(i-1), 2^{-K}i)}(x) \mathcal{N}_m(2^{-K}\lceil 2^K x \rceil)$$

(the expressions $2^{-K}\lceil 2^K x \rceil$ correspond to input quantization masks). Then, it follows from Lemma 1 that

$$\left\| \mathcal{N}_m(x) - \overline{\mathcal{N}_m}(x) \right\|_\infty \leq 2^{-K}.$$

If $k_{\cdot,i'} \neq 0, \sigma(i') \geq t_0$, we denote $\overline{R_K}(f)$ as the expression which we can obtain by replacing $\mathcal{N}_m$ in $\tilde{R}_K(f)$ by $\overline{\mathcal{N}_m}$. Then the following estimation follows:

$$\|\tilde{R}_K(f) - \overline{R_K}(f)\|_r \lesssim 2^{-K}.$$

Consequently, as in the proof of Theorem 1, it follows that

$$R_r(\mathbf{MTN}_{t,2^K}(L, T, E, W, H, S, B), VU_{p,q}^\alpha([0,1]^{dl})) \lesssim 2^{-K} D_{K,dl}^{\eta \vee 1}.$$

For a general $\pi$, we execute the same proof strategy. For each $\Omega_i$, we set $A_i := \{i' | k_{\cdot,i'} \neq 0, \sigma_i(i') \geq t_0\}$. Since $\sigma_i \neq \sigma_{i'}$ for $i \neq i'$ in general, it also holds that $A_i \neq A_{i'}$ for $i \neq i'$ in general. For each $A_i$, we can construct $\left(\overline{R_K}(f)\right)_i$ and this expression $\left(\overline{R_K}(f)\right)_i$ corresponds to each quantization mask.

This completes the proof.

$\square$

