# OpenReview forum: "Approximation ability of Transformer networks for functions with various smoothness of Besov spaces: error analysis and token extraction"
_ICLR.cc/2023/Conference — Submitted to ICLR 2023_

### Official Review · Reviewer_y5B2 · 2022-10-24

**Confidence:** 3
**Correctness:** 4
**Technical Novelty And Significance:** 2
**Empirical Novelty And Significance:** Not applicable
**Recommendation:** 5

**Clarity, Quality, Novelty And Reproducibility:**

The paper is hard to follow, and the implication of the theorems are not clear as the expressions are indirect and hard to parse.

The paper provides new results relating approximation error of Besov spaces using finite-length Transformers, which is timely and possibly important. However, lacks in the discussion of related works, clarity in expressing the final results, and highlighting the novelty w.r.t. Suzuki & Nitanda 2020. These omissions make judging the quality of this work difficult.

The paper seems to be novel as it studies the approximability of Transformers with respect to Anisotropic and Mixed Smooth Besov spaces. However, the technical difficulty in analyzing Transformers with respect to Suzuki & Nitanda 2020 is not discussed clearly.

The paper is reproducible.

**Strength And Weaknesses:**

Strengths:
- The paper studies the approximation error of Transformer networks which is a timely and important problem.
- This paper extends the approximability results of Deep Neural Networks w.r.t. the smooth and anisotropic Besov spaces (Suzuki & Nitanda 2020) to Transformers with finite-length sequences.
- The 'Token extraction property of Transformer networks' in Section 5 highlights some interesting properties which resembles/alludes to attention mechanism in transformers.

Weaknesses:
- The main Theorems are not easy to understand. A corollary where the approximation error is directly linked with the
- The technical novelties as compared to Suzuki & Nitanda 2020 is not discussed clearly. What are the key difficulties faced in extending the results therein to Transformer network? For example, Section 4 seems to follow very similarly as Section 4 of  Suzuki & Nitanda 2020.
- The related work part seems to be lacking. The authors should comment on how the current result improves upon Yun et al. 2020. Furthermore, there has been some other studies after Yun et al. 2020, e.g. Kratsios et al.  "Universal Approximation Under Constraints is Possible with Transformers" ICLR 22. A better placement of this work in the literature is needed.
- The importance of Besov spaces in approximation theory, and its comparison with other concepts, like universal approximation, should be discussed.
- Definition 4 seems to be a restatement of Definition 1.
- Definition 10 is $\pi$ missing from the definition of $VU$?
- "Note that the cardinality of values of masked tokens is finite while the cardinality of values of non-masked tokens are uncountably infinite. Hence, masked tokens have much less information than non-masked tokens." -- I thought fixed finite length input sequences are considered in this paper. Am I missing something here?

**Summary Of The Paper:**

This paper studies how fixed-length Transformer networks can approximate anisotropic Besov and mixed smooth Besov
spaces. They show that the expected approximation error of  fixed-length Transformer networks is mildly dependent on feature dimension $d$ and input length $l$, and is independent of $d$ and $l$ for anisotropic Besov spaces.  Due to the token-wise
parameter sharing property in Transformer network width is independent of the length of the sequence $l$.  They claim superior approximability of Transformer network as compared to Fully connected networks. Moreover, they provide an approximation error bound where the error over $n$ samples decays polynomially with $n$. Finally, by the use of variable smooth Besov spaces the authors try to capture the sparse dependence on the input sequence (e.g. sentence understanding). They show Masked Transformer networks are good at approximating such spaces.

**Summary Of The Review:**

This paper has several shortcomings: placement in literature, clarity in exposition of the results, and clarity in highlighting technical novelties. These shortcomings leave a lot of room for improvement despite the important topic studied in the paper.

---

> ### Author Response · Authors · 2022-11-17
> **Reply from authors (1/2)**
>
> Thank you for your insightful comments. Please look at our responses to your comments below.
>
> (1)
> Q: The main Theorems are not easy to understand.
>
> A:
> We could present our results in a more concise way by moving the precise mathematical details to the appendix and adding more texts in the main text.
> However, in this paper, we have chosen to define various variants of Besov spaces and Transformer network architectures in a mathematically rigorous way, and to state the main results accurately instead of giving a handwavy explanation.
> We think that we can interpret and corroborate the main results obtained this time from a different angle, but in order to do that, it is necessary to significantly rewrite this paper, and there is a high possibility that the point of this paper will also change.
>
> (2)
> Q: The technical novelties as compared to Suzuki & Nitanda 2020 is not discussed clearly. What are the key difficulties faced in extending the results therein to Transformer network?
>
> A:
> The approximation error analyses for the Besov spaces generally utilize the cardinal B-spline approximation theories and then we need to show the cardinal B-spline can be approximated by neural networks.
> The existing work focused on showing that the *fully connected neural networks* can approximate the cardinal B-splines and did not include Transformer networks.
> Indeed, it is far from trivial to show that Transformer networks can approximate the B-splines because Transformer networks are permitted to do limited interactions among tokens which prevents from approximating the product  $ xy $ as Yarotsky did.
> This difficulty is common to previous papers ([1]; [2]). In this paper, by constructing an attention layer which values exchanges between different tokens, we solve the difficulty in limited interactions among tokens. By using attention layers constructed above, we can construct a Transformer network approximate to cardinal B-spline function. Thus, we solve the difficulty in limited interactions among tokens. Moreover, because of parameter sharing between tokens, width of fully connected layers depends only on feature dimension $ d $, not on input sentence length  $ l $.
>
> (3)
> Q: For example, Section 4 seems to follow very similarly as Section 4 of Suzuki & Nitanda 2020.
>
> A:
> The results in Section 4 can be derived from an upper bound of approximation errors and covering numbers of Transformer networks.
> The key difficulties in extending the result therein to Transformer network  on an upper bound of approximation errors are written in the answer above.
> And we discussed calculating covering numbers of Transformer network architectures in Appendix and obtained estimations good enough to accomplish almost minimax optimal rate.
> Thus, although the main result stated in this paper is similar to Suzuki & Nitanda 2020, technical novelties are found in this paper.

---

> > ### Comment · Reviewer_y5B2 · 2022-11-29
> > **Response to Rebuttal**
> >
> > Thanks to the authors for their explanations. The contributions in moving from Fully Connected network to Transformers within the framework of Suzuki & Nitanda 2020, as highlighted in their rebuttal, seems reasonable. The extension to the related work presented in the response is also satisfactory.
> >
> > However, a fair amount of rewriting is needed to integrate the contributions into the main paper. Moreover, going back to point 1, in my opinion, a more accessible statement (either as the main theorem, or as a Corollary) is necessary to understand the approximability of Transformers more clearly.

---

> ### Author Response · Authors · 2022-11-17
> **Reply from authors (2/2)**
>
> (4)
> Q: The related work part seems to be lacking. The authors should comment on how the current result improves upon Yun et al. 2020. Furthermore, there has been some other studies after Yun et al. 2020, e.g. Kratsios et al. "Universal Approximation Under Constraints is Possible with Transformers" ICLR 22. A better placement of this work in the literature is needed.
>
> A:
> We have added some comparisons in page 2.
> Our comparisons with our results and other recent works are as follows:
> [1] proved that Transformer networks are universal approximators of sequence to sequence functions.
> However, since these papers did not assume smoothness of the target function, the results of these papers did not specify an upper bound of Transformer network depths, which corresponds to the fact that universal approximation capability of neural networks did not state anything about an upper bound of the network width. Thus, this paper studies a question which naturally arises as to how properties of the target function are related to the network size and precision required.
> [2] proved that there exists a pair of an input sequence and output particles which minimize a given proper loss functions under a given constraint set. [2] regard an input sentence as a measure, that is, particles or a bag of words, which is an interesting viewpoint.
> However, these papers do not give any quantitative evaluation on the approximation ability of Transformer networks for a given function from an input sequence to an output.   Therefore, these papers' results are completely different from this paper's main purpose to explain why Transformer networks can outperform various NLP tasks represented by target functions in various function spaces.
>
> [1] Yun, Chulhee, Srinadh Bhojanapalli, Ankit Singh Rawat, Sashank J. Reddi, and Sanjiv Kumar. "Are transformers universal approximators of sequence-to-sequence functions?." arXiv preprint arXiv:1912.10077 (2019).
> [2] Kratsios, Anastasis, Behnoosh Zamanlooy, Tianlin Liu, and Ivan Dokmanić. "Universal approximation under constraints is possible with transformers." arXiv preprint arXiv:2110.03303 (2021).
>
> (5)
> Q: The importance of Besov spaces in approximation theory, and its comparison with other concepts, like universal approximation, should be discussed.
>
> A:
> Thank you for your comments. We have added some comments in page 1.
> Comparisons with Besov spaces and Hölder spaces are already contained. Thus, we add a sentence which specifies the importance of Besov spaces: "Besov spaces play an important role in several fields such as wavelet analysis, nonparametric statistical inference and approximation theory."
>
> (6)
> Q: Definition 4 seems to be a restatement of Definition 1.
>
> A:
> We have deleted dupulicate definitions from page 3.
> As you write, Definition 4 states an r-th modulus of smoothness in $ \mathbb{R}^{D} $ and Definition 1 is a special case of Definition 4 when $ D = 1 $ .
>
> (7)
> Q: "Note that the cardinality of values of masked tokens is finite while the cardinality of values of non-masked tokens are uncountably infinite. Hence, masked tokens have much less information than non-masked tokens." -- I thought fixed finite length input sequences are considered in this paper. Am I missing something here?
>
> A:
> We apologize for confusing writings. We have modified this statement in page 9.
> We do not mean that "the cardinality of values of masked tokens" is the token count $ l $, but mean that "the cardinality of values of masked tokens" is "the range of masked token feature values", that is, $ \{ 0, \frac{1}{u}, \frac{2}{u}, \dots, 1 - \frac{1}{u}, 1 \} $.
> We also mean that and the range of non-masked token features is also "the range of non-masked token feature values", that is, a closed interval $ [0, 1] $.
> Thus, the cardinality of the range of masked token feature values is finite while the cardinality of the range of non-masked token feature values are uncountably infinite. Hence, masked tokens have much less information than non-masked tokens.
>
> "Note that the range of masked token feature values is $ \{ 0, \frac{1}{u}, \frac{2}{u}, \dots, 1 - \frac{1}{u}, 1 \} $, while the range of non-masked token features is $ [0, 1] $.
> Thus, the cardinality of the range of masked token feature values is finite while the cardinality of the range of non-masked token feature values are uncountably infinite. Hence, masked tokens have much less information than non-masked tokens."

---

### Official Review · Reviewer_SPZu · 2022-10-24

**Confidence:** 3
**Correctness:** 4
**Technical Novelty And Significance:** 3
**Empirical Novelty And Significance:** Not applicable
**Recommendation:** 5

**Clarity, Quality, Novelty And Reproducibility:**

The clarity of the paper can be improved. The quality and novelty of the paper are high. There is no experimental result in the paper.

**Details Of Ethics Concerns:**

I have no ethics concerns for this paper.

**Strength And Weaknesses:**

**Strong points:**

1. The paper addresses an important issue in the study of transformers, which is the approximation and estimation error of transformers.

2. The paper provides an interesting finding that token-wise parameter sharing in transformers helps reduce the dependence of the network width on the input length.

3. The theoretical results in the paper explain the phenomenon that the transformer models dynamically select tokens to pay attention to.

**Weak points:**

1. Experimental results are needed to corroborate the theoretical results in the paper.

2. The authors do not compare their approximation and estimation error of transformers to the recent results, such as those in [1]. Comparisons to [2] and [3] are mentioned in the paper but need to be elaborated more.

3. The theoretical results in the paper are presented in a confusing way. The impact of the paper and insights from theoretical findings can be clearer if the authors clean up the manuscript and present the results in a more concise way.

**Minor Comments that did not Impact the Score:**

1. Can the results in the paper be extended to a variable length setting and to linear transformers?

**References:**

[1] Kratsios, Anastasis, Behnoosh Zamanlooy, Tianlin Liu, and Ivan Dokmanić. "Universal approximation under constraints is possible with transformers." arXiv preprint arXiv:2110.03303 (2021).

[2] Vuckovic, James, Aristide Baratin, and Remi Tachet des Combes. A mathematical theory of attention. ArXiv preprint arXiv:2007.02876, 2020.

[3] Yun, Chulhee, Srinadh Bhojanapalli, Ankit Singh Rawat, Sashank J. Reddi, and Sanjiv Kumar. "Are transformers universal approximators of sequence-to-sequence functions?." arXiv preprint arXiv:1912.10077 (2019).



**Summary Of The Paper:**

This paper analyzes the approximation and estimation error of transformers to the target functions of fixed-length in a mixed smooth Besov space and in an anisotropic Besov space. The authors also show that transformer models are capable of avoiding the curse of dimensionality and obtaining almost minimax optimal rate. Furthermore, it is pointed out in the paper that token-wise parameter sharing in transformers helps reduce the dependence of the network width on the input length. The authors also show that the transformer models dynamically select tokens to pay attention to.

**Summary Of The Review:**

Overall, this paper could be an interesting theoretical contribution. However, my main concern is the lack of experimental results to validate the theoretical findings in the paper. Also, there is not enough comparison to other approximation and estimation error of transformers from recent works.

Currently, I am leaning toward rejecting the paper. However, given additional clarifications on the two main concerns above in an author response, I would be willing to increase the score.

---

> ### Author Response · Authors · 2022-11-17
> **Reply from authors**
>
> Thank you for your insightful comments. Please look at our responses to your comments below.
>
> (1)
> Q: Experimental results are needed to corroborate the theoretical results in the paper.
>
> A:
> Since this is a highly theoretical paper, we think the theoretical contirubtions are already novel.
> We also believe the assumptions we made in our paper is quite natural because we have seen many situations experimentally where importance varies depending on the features.
> We may conduct some numerical experiments that support our theoretical findings but due to the space limitation, we would like to difer it to the future work.
>
> (2)
> Q: The authors do not compare their approximation and estimation error of transformers to the recent results, such as those in [1]. Comparisons to [2] and [3] are mentioned in the paper but need to be elaborated more.
>
> A:
> Thank you for mentioning your concerns. We have added some comparisons in page 2.
> Our comparisons with our results and other recent works are as follows:
> [1] proved that there exists a pair of an input sequence and output particles which minimize a given proper loss functions under a given constraint set. [2] proved that, when regarding attention layers as functions from measures to measures, attention layers have the Lipschitz continuity property from a viewpoint of Wasserstein distances. Both [1] and [2] regard an input sentence as a measure, that is, particles or a bag of words, which is an interesting viewpoint.
> However, these papers do not give any quantitative evaluation on the approximation ability of Transformer networks for a given function from an input sequence to an output. Therefore, these papers' results are completely different from this paper's main purpose to explain why Transformer networks can outperform various NLP tasks represented by target functions in various function spaces.
> [3] and [4] proved that Transformer networks are universal approximators of sequence to sequence functions.
> However, since these papers did not assume smoothness of the target function, the results of these papers did not specify an upper bound of Transformer network depths, which corresponds to the fact that universal approximation capability of neural networks did not state anything about an upper bound of the network width. Thus, this paper studies a question which naturally arises as to how properties of the target function are related to the network size and precision required.
>
> [1] Kratsios, Anastasis, Behnoosh Zamanlooy, Tianlin Liu, and Ivan Dokmanić. "Universal approximation under constraints is possible with transformers." arXiv preprint arXiv:2110.03303 (2021).
> [2] Vuckovic, James, Aristide Baratin, and Remi Tachet des Combes. A mathematical theory of attention. ArXiv preprint arXiv:2007.02876, 2020.
> [3] Yun, Chulhee, Srinadh Bhojanapalli, Ankit Singh Rawat, Sashank J. Reddi, and Sanjiv Kumar. "Are transformers universal approximators of sequence-to-sequence functions?." arXiv preprint arXiv:1912.10077 (2019).
> [4] Manzil Zaheer, Guru Guruganesh, Kumar Avinava Dubey, Joshua Ainslie, Chris Alberti, Santiago Ontanon, Philip Pham, Anirudh Ravula, Qifan Wang, Li Yang, et al. Big bird: Transformers for longer sequences. In NeurIPS, 2020.
>
> (3)
> Q: The theoretical results in the paper are presented in a confusing way. The impact of the paper and insights from theoretical findings can be clearer if the authors clean up the manuscript and present the results in a more concise way.
>
> A:
> We could present our results in a more concise way by moving the precise mathematical details to the appendix and adding more texts in the main text.
> However, in this paper, we have chosen to define various variants of Besov spaces and Transformer network architectures in a mathematically rigorous way, and to state the main results accurately instead of giving a handwavy explanation.
> We think that we can interpret and corroborate the main results obtained this time from a different angle, but in order to do that, it is necessary to significantly rewrite this paper, and there is a high possibility that the point of this paper will also change.

---

> > ### Comment · Reviewer_SPZu · 2022-11-19
> > **Reply to the Author’s Rebuttal**
> >
> > Thanks the author for the response. Please find below my concerns and questions after reading the author’s rebuttal.
> >
> > 1. The author said, “We also believe the assumptions we made in our paper are quite natural.” Can you please elaborate on this claim more? Which references support this claim?
> >
> > 2. “We may conduct some numerical experiments that support our theoretical findings but due to the space limitation, we would like to defer it to the future work.” There is no limit on the number of pages for the Appendix. It would be more convincing if the author can provide numerical experiments that support the theoretical findings in the paper.
> >
> > Since Discussion Stage 1 is almost ended. You can include the answers and results for my question 1 and 2 without the need to put them in the manuscript. I am looking forward to more discussion in Discussion Stage 2.

---

> > > ### Author Response · Authors · 2022-11-20
> > > **Reply from authors**
> > >
> > > Thank you for your suggestful concerns and questions. Please look at our responses as follows.
> > >
> > > Q.
> > > The author said, “We also believe the assumptions we made in our paper are quite natural.” Can you please elaborate on this claim more? Which references support this claim?
> > >
> > > A.
> > > Since Besov spaces in this paper are used to decide importance of each direction through the smoothness parameter, Besov spaces matches Transformer's noteworthy property of token extraction by the attention mechanism. Hence, the Besov spaces and their variant give natural (and rigorous) mathematical formulation of such a situation where the importance depends on directions (tokes) and depends on input. Indeed, there are many researches which show Transformer networks dynamically change the importance of input tokens, such as [1], [2] and [3].
> > >
> > > [1] Jesse Vig. BertViz: A Tool for Visualizing Multi-Head Self-Attention in the BERT Model. In ICLR Debugging Machine Learning Models Workshop, 2019.
> > > [2] Emily Reif, Ann Yuan, Martin Wattenberg, Fernanda B. Viegas, Andy Coenen, Adam Pearce, Been Kim. Visualizing and Measuring the Geometry of BERT. In NeurIPS, 2019.
> > > [3] Sehoon Kim, Sheng Shen, David Thorsley, Amir Gholami, Woosuk Kwon, Joseph Hassoun, Kurt Keutzer. Learned Token Pruning for Transformers. arXiv:2107.00910, 2022.
> > >
> > > Q.
> > > “We may conduct some numerical experiments that support our theoretical findings but due to the space limitation, we would like to defer it to the future work.” There is no limit on the number of pages for the Appendix. It would be more convincing if the author can provide numerical experiments that support the theoretical findings in the paper.
> > >
> > > A.
> > > Since this result is rigorously proved, our theoretical findings are right unless detects of the proofs in this paper are found.
> > > And, due to the rightness of the result, we think that we may conduct some numerical experiments that support our theoretical findings.
> > > However, as the results presented in this paper are highly theoretical and difficult to show clearly, we would like to defer it to the future work.
> > > Although we cannot add results of additional experiments in this revision, we are going to do additional experiments in the final version to support our theoretical findings.

---

### Official Review · Reviewer_YpHD · 2022-10-25

**Confidence:** 4
**Correctness:** 4
**Technical Novelty And Significance:** 2
**Empirical Novelty And Significance:** Not applicable
**Recommendation:** 5

**Clarity, Quality, Novelty And Reproducibility:**

The paper is well written. The role of transformers is not well explained in the paper. Based on a few papers published by the group of T. Suzuki, this one does not present much originality.

**Strength And Weaknesses:**

Transformer networks are not well understood theoretically. This paper provides some interesting results to verify the efficiency of transformer networks.
Besov spaces are interpolation ones between Sobolev spaces. Once results for Sobolev spaces are established, those for Besov spaces are well expected. This paper considers anisotropic Besov spaces and mixed smooth Besov spaces which are different from the standard Besov spaces and can explain networks' ability of avoiding curse of dimensionality. But the analysis results are not surprising. There have been a few papers published by the group of T. Suzuki. Publishing another one at ICLR would not give too much new insight for transformer networks.

**Summary Of The Paper:**

The authors study approximation and learning of functions from Besov spaces by transformer networks. The results show that transformer networks can avoid curse of dimensionality. Almost minimax optimal rates are presented.

**Summary Of The Review:**

Transformer networks are not well understood theoretically. The topic deserves extensive study.
Besov spaces are interpolation ones between Sobolev spaces. Though anisotropic Besov spaces and mixed smooth Besov spaces are considered in the paper, the analysis is not surprising, based on a few papers published by the group of T. Suzuki.

---

> ### Author Response · Authors · 2022-11-17
> **Reply from authors**
>
> Thank you for your insightful comments. Please look at our feedback below.
>
> (1)
> Q: The role of transformers is not well explained in the paper. Based on a few papers published by the group of T. Suzuki, this one does not present much originality.
>
> A:
> The originality of our contributions can be summarized in the following two points:
> I. The approximation error analyses for the Besov spaces generally utilize the cardinal B-spline approximation theories and then we need to show the cardinal B-spline can be approximated by neural networks.
> The existing work focused on showing that the *fully connected neural networks* can approximate the cardinal B-splines and did not include Transformer networks.
> Indeed, it is far from trivial to show that Transformer networks can approximate the B-splines because Transformer networks are permitted to do limited interactions among tokens which prevents from approximating the product  $ xy $ as Yarotsky did.
> This difficulty is common to previous papers ([1]; [2]). In this paper, by constructing an attention layer which values exchanges between different tokens, we solve the difficulty in limited interactions among tokens. By using attention layers constructed above, we can construct a Transformer network approximate to cardinal B-spline function. Thus, we solve the difficulty in limited interactions among tokens. Moreover, because of parameter sharing between tokens, width of fully connected layers depends only on feature dimension $ d $, not on input sentence length  $ l $.
>
> II.
> We introduce variable mixed smooth Besov spaces and input quantization masks.
> Variable mixed smooth Besov spaces are expressed intuitively: the target function in the variable mixed smooth Besov space changes a direction to
> regard as important or as a noise, according to an input.
> Input quantization masks are used to cut off information of masked tokens to specify that Transformer networks extract much more information from
> non-masked tokens and much less information from masked tokens.
> By introducing these definitions, we showed that transformer networks dynamically select tokens to pay careful attention to.
> This viewpoint is our paper's originality.
>
> [1] Yun, Chulhee, Srinadh Bhojanapalli, Ankit Singh Rawat, Sashank J. Reddi, and Sanjiv Kumar. "Are transformers universal approximators of sequence-to-sequence functions?." arXiv preprint arXiv:1912.10077 (2019).
> [2] Manzil Zaheer, Guru Guruganesh, Kumar Avinava Dubey, Joshua Ainslie, Chris Alberti, Santiago Ontanon, Philip Pham, Anirudh Ravula, Qifan Wang, Li Yang, et al. Big bird: Transformers for longer sequences. In NeurIPS, 2020.

---

### Official Review · Reviewer_hE8B · 2022-11-01

**Confidence:** 3
**Correctness:** 2
**Technical Novelty And Significance:** 3
**Empirical Novelty And Significance:** Not applicable
**Recommendation:** 5

**Clarity, Quality, Novelty And Reproducibility:**

In lines 5-9 of the abstract and the second paragraph of the introduction, the author claimed the transformer could avoid the curse of dimensionality in the setting of fixed-length input and Besov space. This conclusion is mainly from Suzuki (2019) and Suzuki & Nitanda (2021). However, as far as I know, those two papers are not related to the transformer. So in order to claim and convince others that the transformer could also avoid the curse of dimensionality, experiments are necessary. This claim could also be found in the last sentence of page 6, but Suzuki (2019) and Suzuki & Nitanda (2021) never mentioned the transformer in their paper, as far as I know.

In the third paragraph, the author claimed that it is hard to study the representation (approximation) ability of the transformer as tokens interaction is only learned by pairwise dot-products. I agree with this claim. However, many related works in this branch are missed. e.g. Yun et al. (2019), Zaheer et al. (2020), Shi et al. (2021). Indeed, most of them proved that the transformer is the universal approximator of sequence-to-sequence functions with the limited pairwise dot-products, or even in the sparse attention setting. The author should compare this branch of work when analyzing the approximation ability, especially on the proof strategy.



Reference:
Han Shi, Jiahui Gao, Xiaozhe Ren, Hang Xu, Xiaodan Liang, Zhenguo Li, and James T Kwok. Sparsebert: Rethinking the importance analysis in self-attention. In International Conference on Machine Learning, pp. 9547-9557. PMLR, 2021.

Chulhee Yun, Srinadh Bhojanapalli, Ankit Singh Rawat, Sashank J Reddi, and Sanjiv Kumar. Are transformers universal approximators of sequence-to-sequence functions? In ICLR, 2020.

Manzil Zaheer, Guru Guruganesh, Kumar Avinava Dubey, Joshua Ainslie, Chris Alberti, Santiago Ontanon, Philip Pham, Anirudh Ravula, Qifan Wang, Li Yang, et al. Big bird: Transformers for longer sequences. In NeurIPS, 2020.





**Strength And Weaknesses:**

Strength:
The topic is interesting and the proof is detailed: to explain the representation power of the transformer by proving it could avoid the curse of dimensionality, accomplish minimax optimal, decrease the dependence of the network width on the input length, could dynamically select 'focused' tokens.

Weakness:
All claims are not supported by experiments, see details below.

**Summary Of The Paper:**

This paper theoretically analyzed the reason why transformers could outperform fully-connected NNs in NLP tasks.
This paper claimed that the transformer could avoid the curse of dimensionality and accomplish a minimax optimal rate in a setting where the target function takes fixed-length input and belongs to Besov spaces.

**Summary Of The Review:**

The claim is not well-supported due to the lack of experiments and related works.

---

> ### Author Response · Authors · 2022-11-17
> **Reply from authors**
>
> Thank you for your insightful comments. Please look at our responses as follows.
>
> (1)
> Q: In lines 5-9 of the abstract and the second paragraph of the introduction, the author claimed the transformer could avoid the curse of dimensionality in the setting of fixed-length input and Besov space. This conclusion is mainly from Suzuki (2019) and Suzuki & Nitanda (2021). However, as far as I know, those two papers are not related to the transformer. So in order to claim and convince others that the transformer could also avoid the curse of dimensionality, experiments are necessary. This claim could also be found in the last sentence of page 6, but Suzuki (2019) and Suzuki & Nitanda (2021) never mentioned the transformer in their paper, as far as I know.
>
> A:
> There would be large misunderstanding about our contribution.
> Indeed, Suzuki (2019) and Suzuki & Nitanda (2021) do not cover Transformer networks and our explanations about their work clearly state that their analyses are only about the *fully connected network*.
> As you mentioned, the ability of Transformers to avoid the curse of dimensionality can not be shown directly from the existing work.
> Our main contribution is to theoretically prove that; we proved in our paper that the similar results can extend to Transformer networks, which is highly non-trivial.
>
> On the other hand, we would like to note that the *lower bound* of the convergence rate (the minimax optimal rate) shown by Suzuki (2019) and Suzuki & Nitanda (2021) is not dependent on any specific estimator, that is,
> the rate of convergence for any estimator is lower bounded by the minimax optimal rate.
> Our theoretical analysis shows that Transformer networks can also achieve this optimal rate that cannot be improved by any estimators.
>
> (2)
> Q: However, many related works in this branch are missed. e.g. Yun et al. (2019), Zaheer et al. (2020), Shi et al. (2021). Indeed, most of them proved that the transformer is the universal approximator of sequence-to-sequence functions with the limited pairwise dot-products, or even in the sparse attention setting. The author should compare this branch of work when analyzing the approximation ability, especially on the proof strategy.
>
> A:
> We have added some explanations in page 2.
> Comparisons with our results and other recent works can be summarized as follows:
> [1], [2], and [3] proved that Transformer networks is indeed universal approximators. However, they *do not* give any quantitative analysis of the approximation errors.
> That is, what they showed is only the fact that Transformers can approximate any continuous functions but they do not discuss anything about the optimality of the approximation error and estimation errors.
> This point is completely different from ours. In our analysis, we give quantitative analyses of the errors that is characterized by the smoothness parameters of the target function, and we even show their optimality.
> These questions naturally arise as to how properties of the target function are related to the network size and precision required.
>
> [1] Chulhee Yun, Srinadh Bhojanapalli, Ankit Singh Rawat, Sashank J Reddi, and Sanjiv Kumar. Are transformers universal approximators of sequence-to-sequence functions? In ICLR, 2020.
> [2] Manzil Zaheer, Guru Guruganesh, Kumar Avinava Dubey, Joshua Ainslie, Chris Alberti, Santiago Ontanon, Philip Pham, Anirudh Ravula, Qifan Wang, Li Yang, et al. Big bird: Transformers for longer sequences. In NeurIPS, 2020.
> [3] Han Shi, Jiahui Gao, Xiaozhe Ren, Hang Xu, Xiaodan Liang, Zhenguo Li, and James T Kwok. Sparsebert: Rethinking the importance analysis in self-attention. In International Conference on Machine Learning, pp. 9547-9557. PMLR, 2021.

---

### Author Response · Authors · 2022-11-17
**Revision has been uploaded**

Thank you for your careful reading. We have uploaded a revised version.
The main difference from the original one is as follows:

1. We added some comparisons with our results and other recent works.
1. We added a few remarks for the importance of Besov spaces and an explanation about minimax optimal in Remark 2.
1. We have fixed some grammatical errors and typos.

Sincerely yours,
Authors.

---

### Decision · Program_Chairs · 2023-01-20

**Decision:**

Reject

**Justification For Why Not Higher Score:**

- Main reason: Hard to parse results and implications
- Limited development over Suzuki & Nitanda 2021 work

**Justification For Why Not Lower Score:**

N/A

**Metareview: Summary, Strengths And Weaknesses:**

The paper attempts to understand expressive power of popular transformer networks. In particular, the paper studies how fixed-length Transformer networks can approximate anisotropic Besov and mixed smooth Besov spaces. Compared to prior work on expressivity of transformer, this paper provides error rates rather than just stating that transformers are universal approximator (Emphasizing this point in the paper would be beneficial). This problem is of wide interest and the paper could have been timely, but unfortunately the results in the paper are not presented in very accessible manner. Most reviewers found the paper hard to follow and comprehend. The implications of the theorems can be made elaborate by adding a few corollaries of very simple special cases where the expressions are direct and clear to see the benefits. We thank the author and reviewer for engaging in discussion towards making the paper better. It would be useful to add a small discussion on key difficulties in moving from fully connected to transformer network in Suzuki & Nitanda 2021 framework (mostly as described in author response). Incorporating all the reviewer feedback would require a fair amount of rewriting and thus warrant another round of reviewing.

**Summary Of Ac-Reviewer Meeting:**

N/A